# A modelling study of eddy-splitting by an Island/Seamount

Shengmu Yang[1, 2], Jiuxing Xing[1], Daoyi Chen[1, 2], and Shengli Chen[1]

[1]Shenzhen Key Laboratory for Coastal Ocean Dynamic and Environment, Graduate School at Shenzhen, Tsinghua University, Shenzhen 518055, China

[2]School of Environmental Science and Engineering, Tsinghua University, Beijing 100084, China

*Correspondence to*: Daoyi Chen (chen.daoyi@sz.tsinghua.edu.cn)

**Abstract.** A mesoscale eddy's trajectory and its interaction with topography under the planetary β and nonlinear effects in the South China Sea are examined using the MIT General Circulation Model (MITgcm). Warm eddies propagate to the southwest while cold eddies propagate to the northwest. The propagation speed of both warm and cold eddies is

about 2.4 km/day in the model. The eddy trajectory and its structure are affected by an island or a seamount, in particular, under certain conditions, the eddy may split during the interaction with an island/seamount. We focus this research on two parameters R and S (where R and S are two dimensionless parameters of the island size and submergence depth; R is the ratio of the island radius to the eddy radius, and S is the ratio of the seamount submergence depth to the eddy vertical length). The results of sensitivity experiments with varying island or seamount geometry indicate that the eddy

would split in the qualitative range of 1/4<R<2, S<1/5. The scale of the secondary eddy split out decreases as the island diameter or the seamount submergence depth increases. In the splitting process, besides the off-spring eddy, there are also some filaments or eddies with opposite vorticity appearing around the eddy. Eddy-splitting, therefore, is an important way to transform energy from mesoscale to sub-mesoscale in the ocean.

## 1   Introduction

Eddies are common in oceans, both at surface and deep layers, including mesoscale eddies (scale of 100 km) and sub-mesoscale eddies (scale of 10 km) (Itoh et al., 2011; Oey, 2008; Olson et al., 2007). Eddies have gained much attention since they are an important form of material and energy transfer in the ocean (Zhang et al., 2013; Kersalé et al., 2013;

Waite et al., 2007; Zhang et al., 2011; Jacob et al., 2002; Zhang et al., 2014; Wang et al., 2005). Although isolated eddies in open oceans are affected by different factors, many of them have similar kinematic characteristics in general. As many researchers have pointed out, an isolated warm eddy in open oceans moves southwestwards or moves northwards along the western boundary in the northern hemisphere under the planetary β and nonlinear effects (Chang et al., 2012; Wei and Wang, 2009; Nof, 1981; Itoh et al., 2011; Itoh and Sugimoto, 2001; Nan et al., 2011; Cushman-Roisin et al., 1990; Korotaev and Fedotov, 1994). Sutyrin et al. (2003) came to conclusions that β-induced propagation of surface anticyclone drives lower-layer eddies which add a significant southerly component to surface eddy propagation.

The eddy propagation in the ocean is directly affected by topography. The eddy trajectory and structure can be changed due to the interaction with a continental slope, an island or a seamount. The interaction between a warm eddy and a continental shelf slope has been investigated by many researchers based on satellite observations, laboratory and numerical model experiments (Hyun and Hogan, 2008; Rennie et al., 2007; Sutyrin and Grimshaw, 2010; Wei and Wang, 2009; Itoh and Sugimoto, 2001; Smith and O'Brien, 1983). A continental slope is often treated as a wall in the numerical model studies. Previous studies indicate that, the eddy-wall collision can cause the eddy leaking water along the wall and generates along-wall jets which can be related to nonlinear Kelvin waves (Nof, 1988; Shi and Nof, 1994; Reznik and Sutyrin, 2005). When a patch of fast moving water catches up with a slower one, an eddy could be generated near the nose of the along-wall jet (Stern, 1986, 2010). Besides the jets and eddies, during the evolution of an isolated eddy near a wall, nonlinear Kelvin waves can be excited due to the geostrophic adjustment which can trap and transform water along the wall (Umatani and Yamagata, 1987; Dorofeyev and Larichev, 1992). In contrast to the case with a continental slope, when eddies encounter an island or seamount, the eddy could split into two eddies because of the erosion by the isolated topography (Herbette et al., 2003; Simmons and Nof, 2002; Dewar, 2002; Herbette et al., 2005; Luo and Liu, 2006; Cenedese, 2002).

Simmons and Nof (2000) obtained the essential conditions for a barotropic eddy splitting by using a wall moving into the eddy: even for infinitesimal splitting, which arises from weak collisions, the wall length must be at least a radius of the eddy. Drijfhout (2003) discussed the anticyclonic eddy splitting mechanism which is that anticyclones cannot split by barotropic processes alone, and baroclinic instability is a necessary ingredient for splitting to occur. Using an

isopycnal ocean circulation model, Herbette (2003) analyzed the behavior of a surface-intensified anticyclonic eddy encountering an isolated seamount, and the erosion often results in a subdivision of the eddy. Wang and Dewar (2003) studied the meddy–seamount interaction. The initial meddy splits into two meddies in their experiments, but meddies are able to survive as coherent vortices because of strong potential vorticity anomalies. Numerical estimates of the transformed eddy structure indicate that topographic interactions provide powerful mechanisms for the baroclinic eddy evolution (Sutyrin et al., 2011).

There are plenty of mesoscale and sub-mesoscale eddies existing in the South China Sea (SCS), and most of them propagate to the southwest (Chang et al., 2012; Nan et al., 2011). In particular, mesoscale eddies occur frequently in the northern SCS (Hwang and Chen, 2000; Chang et al., 2012; Zhang et al., 2013; Nan et al., 2011; Wang et al., 2003; Wang et al., 2005), and the number of cold eddies is similar to that of warm eddies. Therefore, it is of importance to find out the difference between the cold and warm eddies.

Furthermore, the SCS is populated with numerous islands and seamounts. Therefore, most eddies are affected by the topography variation in their movement. The change of eddy structure over topography has an important influence on its dynamics, while it is an important mean of energy transfer among different scales and affects the coastal ocean environment (Kersalé et al., 2013; Drijfhout, 2003; Dunphy and Lamb, 2014). Chang et al. (2012) found from satellite observations that an anticyclonic eddy (warm eddy) with a diameter of 120 km was split by the Dongsha Atoll situated on the slope in the northern SCS. Because of difficulties in catching the entire process of eddy splitting by both satellite observations and situ measurements, there are few cases of eddy-island interactions found by satellite images so far. Particularly, the phenomenon of eddy-splitting reported in Dongsha SCS lacks sufficient measured data to systematically describe the process of splitting (Chang et al., 2012). In addition, eddies may split during interaction with a curved continental slope. Kersalé et al. (2013) investigated a coastal anticyclonic eddy in the western part of the Gulf of Lion in the northwestern Mediterranean Sea, where eddies split in a similar pattern as in the case of Dongsha Atoll. This provides a wider application prospect for any eddy-splitting role in the interaction with topography. However, it is not clear whether an eddy can always be split by an island/seamount and how the scale of the isolated topography influences the eddy eddy-splitting. Recently, Li et al. (2016) used the Genealogical Evolution Model (GEM) to track the dynamic

evolution of mesoscale eddies in the ocean. They can distinguish between different dynamic processes including merging and splitting, but the special processes and characteristics of eddy splitting by an island have not been elucidated completely.

In this study, we constructed an idealized eddy in a numerical model according to the features of the observed eddies in the SCS to examine its kinematic characters and test eddy splitting process using numerical simulations. Moreover, inspired by the eddy splitting near the Dongsha Island in the SCS, we vary the island size and seamount submergence depth to investigate the influence of the island on the eddy, and then to analyse the effect of the island and the seamount on the mesoscale eddy evolution (weakening and destruction) as the eddy approaches the obstacles.

This paper is organized as follows: Sect.2 describes the eddy structure used in the model and the method of eddy identification. Sect.3 introduces the model. The model results, including a comparison of eddy trajectories between the warm eddy and cool eddy, and the effect of an island and seamount on eddy deformation will be presented in Sect.4. A summary and discussion is given in Sect.5.

## 2   An idealized mesoscale eddy

### 2.1 The eddy structure

An idealized mesoscale eddy is initialized with an axisymmetric Gaussian-type profile based on long term moored observations (Zhang et al., 2013), Argo float data and the merged data products of satellite altimeters (Chen et al., 2010). Temperature profiles from observations are fitted into an equation of

$$T(z) = T_b(z) + a_z e^{-\frac{x^2+y^2}{2L^2}} \qquad\qquad (1)$$

where $T_b(z)$ is background temperature; $a_z$ is a function parameter varying with depth (z) and L is constant $1.5 \times 10^4$ m; x, y and z are position coordinates.

The eddy's initial velocity is calculated using the thermal wind balance with zero velocity at the ocean bottom. The density distribution is obtained from a state equation according to Jacket and Mcdougall (1995). Fig.1 shows the temperature and azimuthal velocity distribution on the cross section through the eddy center. The initial eddy is 60 km

in diameter and 500 m in depth with a total water depth of 2000 m. The maximum surface velocity is about 0.9 m/s, and the maximum surface elevation is 0.5m.

## 2.2 Eddy identification and definition of the eddy boundary

There are different methods to identify an eddy and here we use the Okubo-Weiss method (Okubo, 1970; Weiss, 1991) to identify the eddy that we constructed in the model and define the boundary of the eddy. The Okubo-Weiss parameter W is given by

$$W = s_n{}^2 + s_s{}^2 - \omega^2 \qquad (2)$$

$$\omega = \frac{\partial v}{\partial x} - \frac{\partial u}{\partial y} \; ; \; s_n = \frac{\partial u}{\partial x} - \frac{\partial v}{\partial y} \; ; \; s_s = \frac{\partial v}{\partial x} + \frac{\partial u}{\partial y} \qquad (3)$$

where $\omega$ is the vertical component of relative vorticity; $s_n$ and $s_s$ represent the strain and shear deformation; and u and v are eastward and northward velocities respectively.

Because the velocity field within an eddy is dominated by its rotation, ocean eddies are generally characterized by negative values of W. In this study, we use $W < -0.2\sigma_w$ to define the core region of the eddy, where $\sigma_w$ is the standard deviation of W in the study region. This way to identify an eddy has a tendency toward excess of eddy detection (Doglioli et al., 2007), so we combine the potential vorticity anomaly distribution, velocity field and temperature anomaly to determine the main eddy that we focus on, and ignore the smaller circulations due to the eddy-topography interaction.

## 3    Numerical model and initialization

The MITgcm (**MIT G**eneral **C**irculation **M**odel) (Adcroft et al., 2011) is used in this study. Its non-hydrostatic formulation enables us to simulate fluid phenomena over a wide range of scales. However, we only use the hydrostatic form of the model as we expect that the non-hydrostatic dynamics plays minor roles in our problem (to capture the non-hydrostatic dynamics we would have to use much finer resolution than used here). The model domain is 500 $\times$ 450 km$^2$, and the depth of the ocean used in the model is 2000 m. The horizontal resolution is 2.5 km; in the vertical, 28 levels are used with 50 m resolution in the upper 1000 m and the resolution gradually coarsens in the lower 1000m. The

Coriolis parameter $f = 9 \times 10^{-5}$ s$^{-1}$ and the planetary parameter $\beta = 2 \times 10^{-11}$ m$^{-1}$s$^{-1}$ (here the main reason to use $\beta$ plane rather than f plane is the $\beta$ effect being the main force for the movement of an eddy, see Sect.4). The model boundaries all are open, and the Orlanski radiation condition is used. In the horizontal, we use Smagorinsky viscosity with a parameter of 0.2. In the vertical, the eddy viscosity is $5.0 \times 10^{-4}$ m$^2$/s. For the temperature equation, the vertical eddy diffusivity is $10^{-4}$ m$^2$/s and horizontal eddy diffusivity is set to zero.

In the model, both the warm eddy and the cold eddy are initialized with an axisymmetric Gaussian-type profile described in Sect.2. The temperature decreases with the depth in the upper 1000 m and is set to a constant value of 4℃ below 1000 m. A constant salinity of 35 psu is used which does not affect the model results.

For the model with flat topography, the eddy is located at the center of the model domain to test the difference between warm- and cold-eddy trajectories. In the cases studying the interaction between an eddy and an island/seamount, the island/seamount with different sizes/depths is located in the central path of the eddy, and all islands and seamounts are cylinder shaped. We run the model from the initial state of rest for 50 days in order to compare different effects of obstacles on eddies.

## 4    Results

Our main attention is on the eddy-splitting due to the interaction between eddies and obstacles, and a series of experiments based on the idealized eddy structure in the SCS have been carried out (Table 1.). The eddy diameter is 60 km, and the initial location of the eddy center is x=250 km, y=225 km.

We first examine the eddy trajectories and its characteristics without any island/seamount. Then we focus on the interaction between the eddy and the island/seamount, and the sensitivity of eddy-splitting to the island size and seamount depth.

### 4.1 The trajectories of warm and cold eddies

In our first set of numerical experiments, an eddy (warm or cold) is located at the center (x=250 km, y= 225 km) of the domain with open boundaries and a flat bottom (Fig.2). When the eddy is a warm eddy (anticyclonic eddy in the northern

hemisphere), it moves towards the southwest direction in a flat bottom ocean. At the beginning of the model integration, the eddy will adjust itself to a dynamic balance. As a result, the speed of the eddy movement is relatively small. After the model reaches its balance, the speed of the eddy increases, to a constant value of 2.4 km/day after 40 days. The speed of the warm eddy in the model is similar to that of the study of (Wei and Wang, 2009). The eddy propagation speed is

influenced by the eddy size and the β effect, which is a function of the local latitude. Therefore at a different latitude, the eddy has different speed. Fig.3 shows that the average speed over 50 days of the warm eddy is 1.75 km/day which is smaller than the eddy speed in the natural conditions in the SCS region because of the adjustment in the early stage of the model run.

With a cold eddy (cyclonic eddy in the northern hemisphere) in the same situation, the movement direction is northwest

under β and nonlinear effects. The speed increases from the beginning of the model integration which is the same as for the warm eddy. The speed reaches a constant value of 2.3 km/day after 40 days. From the trajectory and speed variation during the eddy movement, we can see that the warm and the cool eddy have similar kinematic characteristics. However, the cold eddy moves to a higher latitude in the northern hemisphere while the warm eddy moves to a lower latitude because of their different spin directions.

When an isolated eddy propagates in open oceans with a flat bottom, while the β effect drifts the eddy westwards (Shi and Nof, 1994), nonlinearity provides the meridional component of movement (e.g. Chang et al., 2012; Hyun and Hogan, 2008). From the results of the model, the warm eddy generally moves in the southwest direction in the northern hemisphere which agrees with previous studies. The trajectory of the cold eddy is mirror symmetric with the warm eddy (Fig.2). Both eddies' propagation speed is about 2.4 km/day which is smaller than the value in previous numerical

investigations which used mesoscale eddies with 100 km horizontal scale (Wei and Wang, 2009; Sutyrin et al., 2003). The eddy propagation speed associated with the eddy size and the propagation speed increases with increasing eddy size but will be limited by the maximum Rossby wave phase speed.

## 4.2 Eddy-splitting

The influence of an island on the eddy deformation is explored in this study. According to the eddy-splitting at Dongsha

Island in the SCS (Chang et al., 2012), we set an island on the path of the warm eddy based on the first case we have examined. The diameter of the island is 20 km. At the beginning of the model integration, the eddy is not influenced by the island because the distance between the eddy and the island is not sufficiently close. As the eddy moves towards the island along its trajectory, the eddy eventually interacts with the island.

It is evident from Fig. 4 when the eddy collides with the island, there is another weak warm eddy formed on the other side of the island. The two eddies have similar diameters, but the secondary eddy is weaker than the main one which can be seen from SSH (Sea Surface Height), temperature, PVA (Potential Vorticity Anomaly), and O-W field (Fig.5). In eddy-splitting, the temperature and PVA can be seen as a tracer. From the temperature distribution we can find that the water of the secondary eddy is derived from the original eddy, so we believe that the secondary eddy comes from eddy-

splitting rather than being formed independently. After the eddy-splitting, the two eddies move away from the island along their own trajectories as independent eddies. When a cold eddy encounters an island with 20 km diameter in its trajectory, the eddy split in the same way with the warm eddy (Fig.6). Therefore, only the warm eddies are used to study the influence of an island/seamount on the eddy-splitting.

As mentioned previously, when an eddy collides with an island, the eddy can split into two eddies with similar rotation

characters. Here we examine the evolution of the eddy-splitting process. Fig.7 shows the temperature field evolution of an anticyclonic eddy colliding with an island with a diameter of 20 km. The eddy is initially located at 40 km away northeast to the island. Then the eddy moves towards the island at 0.023 m/s. At t=20 days, the eddy gradually collides with the island. The isolated anticyclone is cut by the island. The fluid at the edge leaks to right (looking off-shore) due to the presence of the solid boundary of the island. The eddy loses mass along the edge of the island, creating a jet

moving away from the eddy.

As the inertia and β effect push the eddy continually closer to the island, more and more warm water leaks to form a jet with higher velocity. From t=26 days, because of the curve edge of the island, the jet moves forward off the boundary. The jet trajectory curves to the right side under the influence of the earth rotation. Until t=32 days, the water leaking as a jet become weaker as the eddy stop squeezing to the island. At the same time, the warm water trapped by the jet gathers

at the downstream and merges into the newly formed anticyclone eddy.

The radius of the newly formed anticyclone is about 25 km which is similar with the parent eddy, but its strength is weaker. Under the boundary effect, both two eddies move away from the island. As a result, the parent anticyclonic eddy splits into two anticyclonic eddies during the interaction with the island.

For better understanding the mechanism of the eddy-splitting process, the PVA field is analyzed which is shown in Fig.8. The eddy is composed by two parts: ones is the inner part with negative PVA; the other is the outer annulus with positive PVA. As shown in the figure, from t=22 days, at the start stage, the water leaked out is outer annulus water with positive PVA and forms the origin jet. When the jet flows off the boundary from t=24 days, there is an anticyclonic eddy formed due to the flow shear effect at the corner which is the separation point of the jet and the boundary. At t=32 days, as the eddy pushes closer to the island, more warm water with lower vorticity flows into the newly formed anticyclone under the influence of the Coriolis force. When the warm water merged into the anticyclonic eddy, the new anticyclone matures gradually similar to the parent eddy by geostrophic adjustment and moves off the island. As shown in Fig.8, the newly formed anticyclonic eddy is weaker than its parent eddy counterpart.

The position of the newly formed anticyclone is controlled by the separation point of the jet and the island boundary, and therefore is influenced by the boundary curvature which is a function of the island scale. As the island scale increases, the azimuthal angle (clockwise is positive) of the new anticyclonic eddy to the parent eddy decreases. The relationship of the positions of the eddies and the island will be discussed in section 4.3.

When the eddy encounters an obstacle, the trajectories and the speed are usually drastically altered. The results show that the speed of the eddy decreases significantly when the eddy interacts with the island. Shi and Nof (1994) pointed out that the image effect and the rocket effect (caused by the jet) usually dominate when colliding with a solid obstacle, and the effect would change the original movement trend combined with the boundary effect. At the same time, the generation of a weak cyclonic eddy during the interaction of warm eddies with an island/seamount adds a significant effect on the eddy propagation.

Actually, an anticyclonic eddy can never split on its own. Applying the conservation law of integrated angular momentum (IAM), Nof (1990) demonstrated this. As a result, when a warm eddy splits, the IAM has to increase as the newly formed eddies move away from their original center. When a warm eddy is forced by the solid boundary of an island or a

seamount, in the lower layer there has to be a transfer of IAM from the surrounding fluid to the core region of the eddy (Drijfhout, 2003). In order to show the change of integrated angular momentum before and after the eddy interacts with the island, the PVA at surface layer (depth=100 m) and deep layer (depth=1000 m) at t=30 days and t=50days are shown in Fig.9. Compared with the PVA field at t=30 days, the maximum upper anticyclonic PVA decreases at t=50 days because of the splitting while maximum lower cyclonic PVA increases.

## 4.3 The effect of island sizes on eddy-splitting

Observational data, including satellite images and situ measurements, indicates that when an eddy collides with a continental slope or a small island, there is no eddy-splitting, only changes of its trajectory (Jacob et al., 2002; Nan et al., 2011; Wei and Wang, 2009). In order to find out the parameter ranges of eddy-splitting, we use a series of islands with different diameters at the same location in the model. Before that, interactions of different sized islands and eddies were investigated. Take, for example, the eddies with 90 km ($Eddy_{90}$) and 60 km ($Eddy_{60}$) in diameter, the eddy-splitting pattern of $Eddy_{90}$ interacting with an island of 120 km diameter is similar to that of $Eddy_{60}$ interacting with an island of 90 km (Fig.10). Although the islands and eddies all are different in size in comparison, they have an approximately same ratio of the island radius to the eddy radius in each experiment.

We, therefore, define two dimensionless parameters R and S to represent the size and submergence depth of an obstacle, namely,

$$R = \frac{R_{ob}}{R_{ed}} \tag{4}$$

$$S = \frac{D_{sb}}{D_{ed}} \tag{5}$$

where $R_{ob}$ is the radius of an obstacle; $R_{ed}$ is the radius of an eddy; $D_{sb}$ is the seamount submergence depth and $D_{ed}$ is the vertical extent of an eddy.

The eddy collides with the islands in 20 days and interacts with them as we have described previously. Fig.11 shows when the island is small enough, namely R < 1/4, the eddy does not split. Instead, the eddy will move through the obstacle, although the eddy structure deforms during the interaction process, and then recovers back after the interaction. As the

island diameter increases, the 'passing through' eddy gradually turns to splitting as a result of the eddy-island interaction. The eddy-splitting happens in the parameter range of 1/4 <R< 2. As the island diameter increases to R>2, a filament splits out from the eddy. This phenomenon is not considered as eddy-splitting in this study. In the last example, when the eddy collides with a solid wall (which can be seen as an island with an infinite diameter), the eddy propagates to the higher latitude along the boundary which agrees with previous studies (Wei and Wang, 2009).

From the eddy-splitting processes with different sizes of islands, we can find that the locations of the secondary eddy split out are related to the island size. Fig.12 shows the position relationship of the two eddies and the island. The angle ($\theta$) between the secondary eddy and the position of collisional origin varies with the different island sizes (R). The distribution of the angle ($\theta$) and the island size (R) are shown in Fig.13. The fitting curve demonstrates that the empirical relation between the angle and the island size can be written by

$$\theta \sim f(R) = 2.6R^{-0.663} \tag{6}$$

where $f(R)$ is the angle (rad) between the two split eddies related to the island.

## 4.4 Effect of a seamount on eddy-splitting

In natural oceans, islands are just part of the topography and there are more seamounts which are submerged under the sea surface. The effect of seamounts on ocean dynamics is different from that of islands. The submergence depth and the size of a seamount are key factors in the eddy-splitting. During the interaction between an eddy and a seamount, the lower part of the eddy is affected directly by the solid seamount while the upper part is not, then the vertical structure of the eddy is deformed significantly. As a result, its trajectory and splitting process is different from that of the interaction between an eddy and an island.

4.4.1 The effect of the seamount submergence depth

Here we investigate the effect of a seamount submergence depth on eddy-splitting. The experiments are set up based on the cases of R=1/4, 1, 2, which have typical eddy splitting. Model results for the seamount with a diameter of 60 km are presented in Fig.14. When the submergence depth is 50 m, which is shallow, the interaction process between the eddy and the seamount is similar to that of the interaction between an eddy and an island. With the increase of depth, eddy-

splitting becomes weaker and weaker. When the seamount submergence depth is 100 m, the upper layer of the eddy moves under the inertia effect while the lower part is hindered by the seamount; this leads to the change of eddy vertical structure and the upper water of the eddy is stranded by the seamount. At the same time, the filament, which sheds from the eddy, is closer to the main body of the eddy compared with the case of an island.

When the seamount submergence depth is 200 m, the effect of the seamount on the eddy structure has weakened greatly compared with the seamount submergence depth of 100 m. Apart from the filament shedding, there is no significant change in the main structure of the eddy. The result also shows that the seamount with S=2/5 cannot induce the eddy-splitting. When the submergence depth is 500 m (S ≈1), the seamount only affects the bottom of the eddy. The eddy trajectory changes under this circumstance. Fig.14 (e) shows that the eddy will bypass the obstacle from the left side under the effect of the secondary circulation in the deep layer. When the seamount submergence depth is 1000 m (S>1), the existence of the seamount does not impact the eddy motion, and the warm eddy moves toward southwest which is similar to the case of a flat bottom.

From the results of the numerical experiments, we find that eddy-splitting happens roughly in the range of S <1/5 when the seamount diameter is 60 km. Similarly, when the seamount is 10km in diameter, the eddy-splitting occurs at S <1/10. Actually, the range of eddy-splitting in the seamount cases is related to the seamount horizontal size as discussed in the next part.

4.4.2 The effect of the seamount size

When an eddy collides with a seamount, the effect of the seamount on eddy-splitting is weaker than that of an island. The effect of the seamount on eddy-splitting is not only determined by the submergence depth but also influenced by the seamount horizontal scale. Here we test three different sized seamounts with the same submergence depth (Fig.15). During the interaction between the eddy and the seamount with 15 km diameter, the eddy does not split, and when the seamount diameter is 60 km, a small eddy is split out while the main eddy deforms. For the seamount with 120 km diameter, intense deformation occurs to the eddy without splitting.

For a seamount, the eddy-splitting happens in a narrower band of horizontal scale compared with an island. As the seamount submergence depth increases, the influence of the seamount on eddy deformation decreases. Therefore the

band of seamount horizontal scale for which the eddy-splitting occurs becomes narrower and narrower as the submergence depth increases.

Concerning eddy evolution in the ocean, we have explored the effect of topography such as islands and seamounts on eddy-splitting. According to the results we obtained, the dependence of eddy-splitting on the parameters R and S is

summarized in Fig.16. This diagram illustrates the main settings of the experiments and the red area is where eddy-splitting occurs.

## 5    Summary and discussion

Motivated by the eddy-splitting near Dongsha Island in the SCS, we have explored the eddy's trajectory and effect of topography on an idealized eddy evolution. MITgcm is used in the study of the effect of topography on eddy evolution

including eddy trajectory and its structure, particularly the eddy-splitting when the eddy collides with an island/seamount. The topography used in the numerical experiments includes a flat bottom, islands with different diameters and seamounts with different submergence depth. Eddies colliding with the topography all have the same initial structure. The simulation results of PVA, SSH, temperature and O-W parameter are analyzed.

The model eddies (both warm and cold) move at a speed of 2.4 km/day in open oceans under the planetary β and nonlinear

effects. The warm (cold) eddy moves southwestward (northwestward). The eddy speed and trajectory are influenced by topography. Generally speaking, the effect of topography starts when the eddy is some distance away from the island. The island leads to the eddy's trajectory changing and slows down the movement of the eddy. Because of the inertia of the eddy movement, eddies interact with obstacles by the collision. The dependence of eddy behaviors on the horizontal scale and submergence depth of an obstacle can be summarized using two dimensionless parameters R and S. We have

shown the qualitative range of eddy-splitting using the results of numerical model experiments. During the eddy-splitting, the location of a secondary eddy detached from the main eddy is related to the size of the island or the seamount. Results of the model experiments show that the relationship between the angle of two eddy directions f(R) and the dimensionless parameter R can be written as  $f(R) = 2.6R^{-0.663}$ .

Because observational data of eddy-splitting in oceans is scarce, we need more and comprehensive measurement data in

combination with numerical models to explore the dynamic mechanism of eddy-splitting further. In addition to the dimensionless parameters R and S, there are other physical effects and control parameters in eddy-splitting such as the strength of an eddy which depends on the stratification (Thiem et al., 2006), and the movement speed of the eddy. In this paper, a single eddy interacting with an island or seamount was studied. However, there may be another scenario that a sequence of eddies hits an island. The result of the first eddy interacting with the island may be different from that of the eddy behind. In our study, the island is placed in the middle of the trajectory of the eddy. The results can be much more complicated when eddies hit more to one side of the island. In short, the eddy-topography interaction is a systematic and complex problem. In order to better understand the issue, many involved factors need to be explored. Meanwhile, an investigation using more realistic model settings, such as the real topography, density stratification and forcing of the northern SCS is in progress.

**Acknowledgements**

The authors would like to express their sincere gratitude to the insightful comments from Prof. J Huthnance of NOC (UK). The very constructive comments from the editor (Eric J.M. Delhez) and referees, in particular, Dr. Y. Lu (Bedford Institute of Oceanography, Fisheries and Oceans Canada), Prof. J. Berntsen (University of Bergen) and an anonymous referee have greatly helped to improve the manuscript. This work was supported by the National Key Basic Research Program of China (Program 973) (grant 2014CB745001), the Environmental Protection Special Funds for Public Welfare (201309006), the Shenzhen Special Funds for Future Industry Development (201411201645511650) and S. Chen is supported by the China Postdoctoral Science Foundation (2016M591159).

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

**Tables**

Table 1. List of different topography used in the experiments

| Case | Type | Diameter | Submerge Depth | Center Location | Outcome |
|---|---|---|---|---|---|
| 1 | flat | - | - | - | No Splitting |
| 2 | Island | 10 km | 0 | (213 km, 176 km) | No Splitting |
| 3 | Island | 15 km | 0 | (211 km, 174 km) | Split |
| 4 | Island | 25 km | 0 | (207 km, 170 km) | Split |
| 5 | Island | 60 km | 0 | (195 km, 158 km) | Split |
| 6 | Island | 90 km | 0 | (184 km, 147 km) | Weak Splitting |
| 7 | Island | 120 km | 0 | (173 km, 136 km) | Weak Splitting |
| 8 | Island | 150 km | 0 | (162 km, 125 km) | Weak Splitting |
| 9 | Island | 300 km | 0 | (109 km, 72 km) | Filament |
| 10 | Island | Infinite | 0 | - | Filament |
| 11 | Seamount | 15 km | 50 | (211 km, 174 km) | No Splitting |
| 12 | Seamount | 15 km | 80 | (211 km, 174 km) | No Splitting |
| 13 | Seamount | 15 km | 100 | (211 km, 174 km) | No Splitting |
| 14 | Seamount | 60 km | 50 | (195 km, 158 km) | split |
| 15 | Seamount | 60 km | 80 | (195 km, 158 km) | split |
| 16 | Seamount | 60 km | 100 | (195 km, 158 km) | Weak Splitting |
| 17 | Seamount | 60 km | 200 | (195 km, 158 km) | Filament |
| 18 | Seamount | 60 km | 500 | (195 km, 158 km) | No Splitting |
| 19 | Seamount | 60 km | 1000 | (195 km, 158 km) | No Splitting |
| 20 | Seamount | 90 km | 50 | (184 km, 147 km) | split |
| 21 | Seamount | 120 km | 100 | (173 km, 136 km) | No Splitting |
| 22 | Seamount | 120 km | 150 | (173 km, 136 km) | No Splitting |
| 23 | Seamount | 150 km | 100 | (162 km, 125 km) | No Splitting |

**Figures**

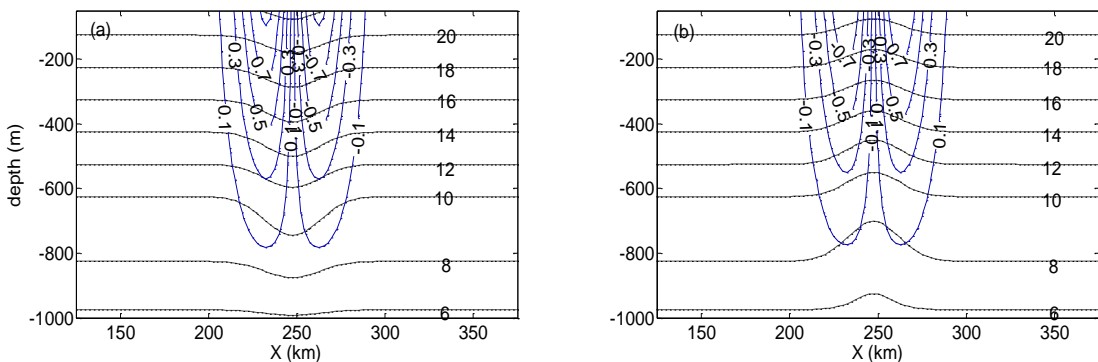

Figure 1. Initial velocity (m/s) and temperature (℃) profiles of the model warm eddy (a) and cold eddy (b).

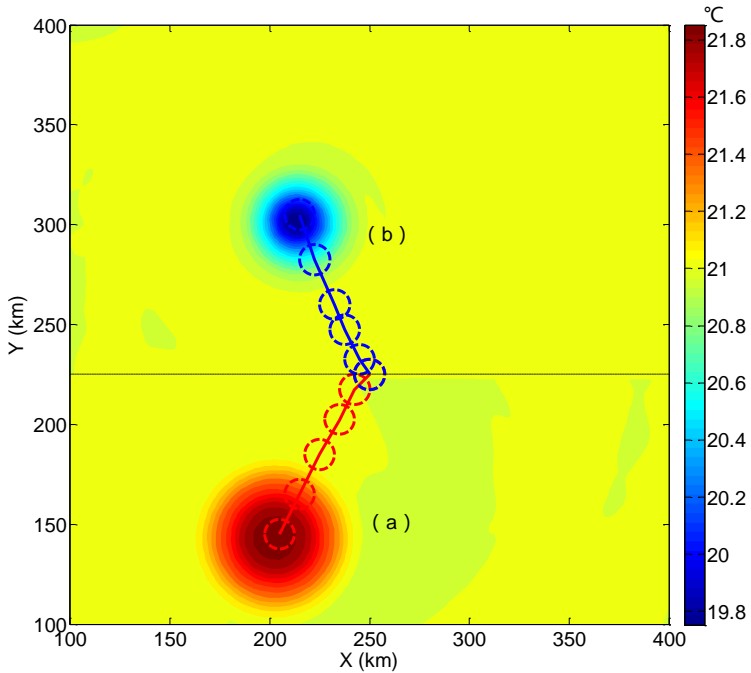

Figure 2. Eddy trajectory over a flat bottom: (a) warm eddy; (b) cold eddy) for 50 days. The temperature field shown in colours is a snapshot of the eddy at t=50 days at 100 m depth. The trajectory of the eddy center is depicted by circles every 10 days.

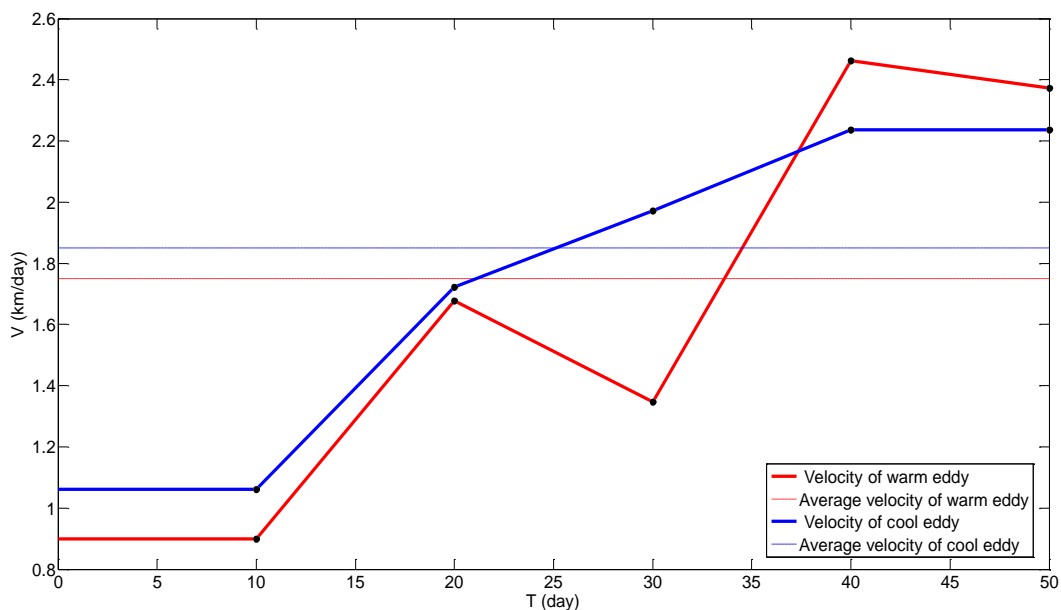

Figure 3. The speed of eddies over a flat bottom ocean. Solid lines: time series of speed; dashed lines: the speed averaged over 50 days.

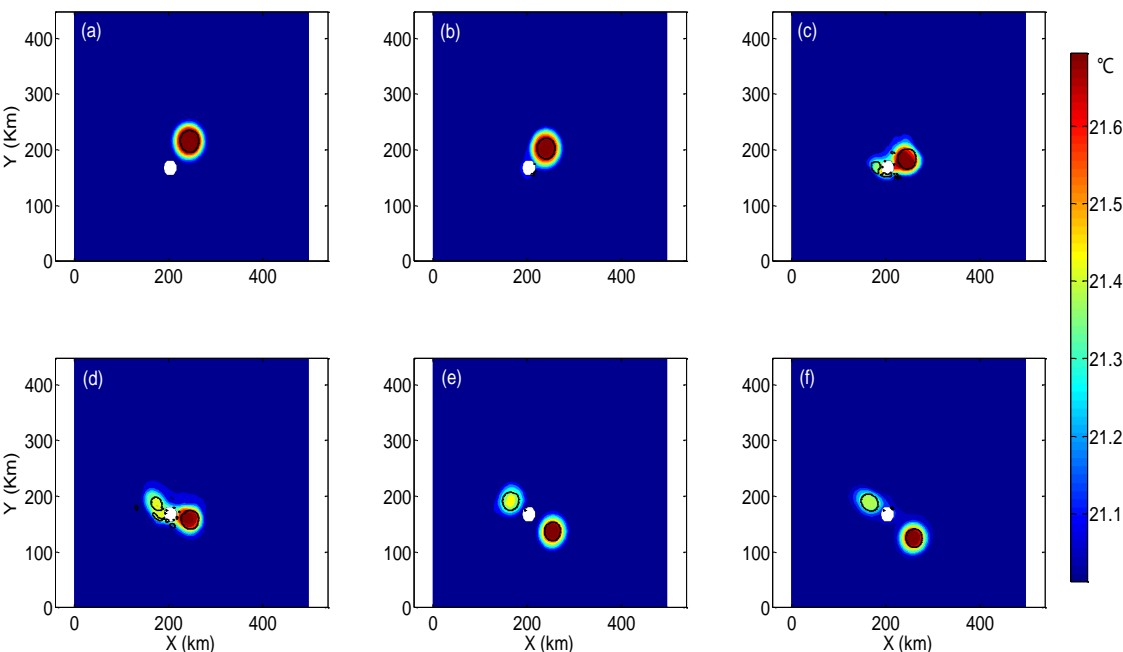

Figure 4. The process of eddy-splitting induced by the interaction with an island of 20 km in diameter over 50 days. A time series of snapshots of temperature at 100 m depth is shown in colours. (a): the initial state; (b): 10 days; (c): 20 days; (d): 30 days; (e): 40 days; (f):50 days. The black solid lines are O-W parameter with value of $-0.2\sigma_w$.

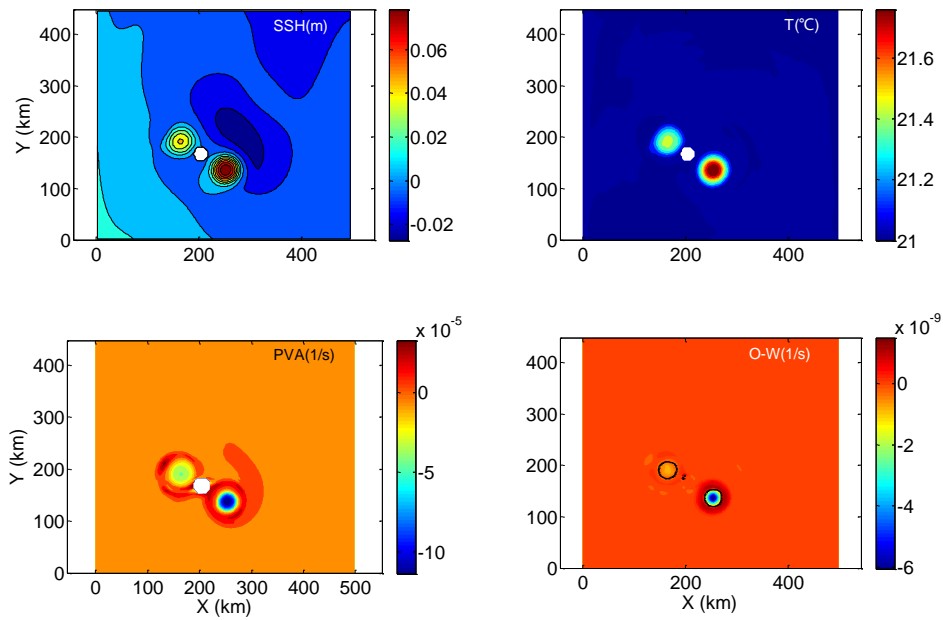

Figure 5. The warm eddy splits into two eddies during the interaction with an island of 20 km in diameter on day 50 (the results shown is at 100 m depth).

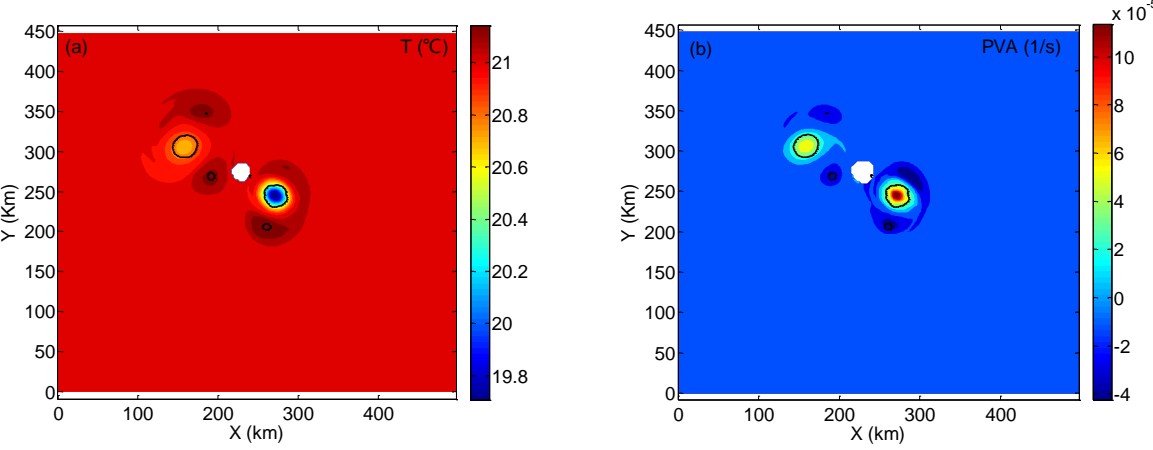

Figure 6. The cold eddy splits into two eddies during the interaction with an island of 20 km in diameter on day 50 (the temperature and PVA shown are at 100 m depth). The black solid lines are O-W parameter with value of $-0.2\sigma_w$.

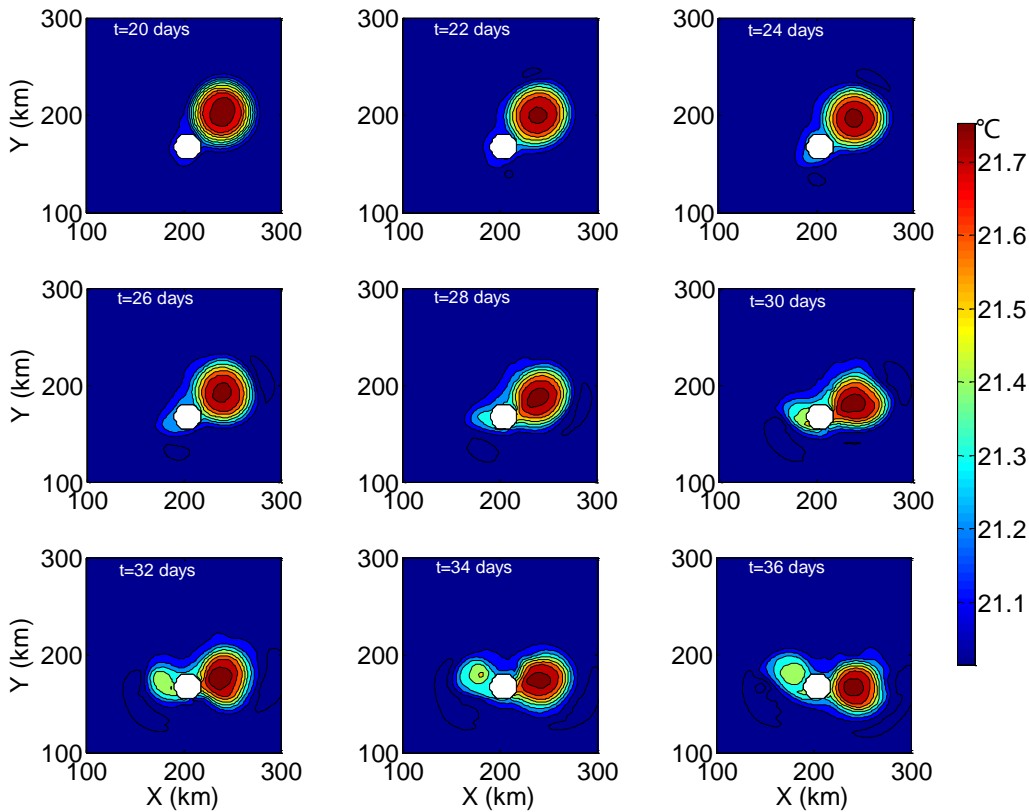

Figure 7. The temporal evolution of the eddy in the interaction with an island of 20 km in diameter. The colours represent the temperature at 100 m depth and the solid lines are temperature contours.

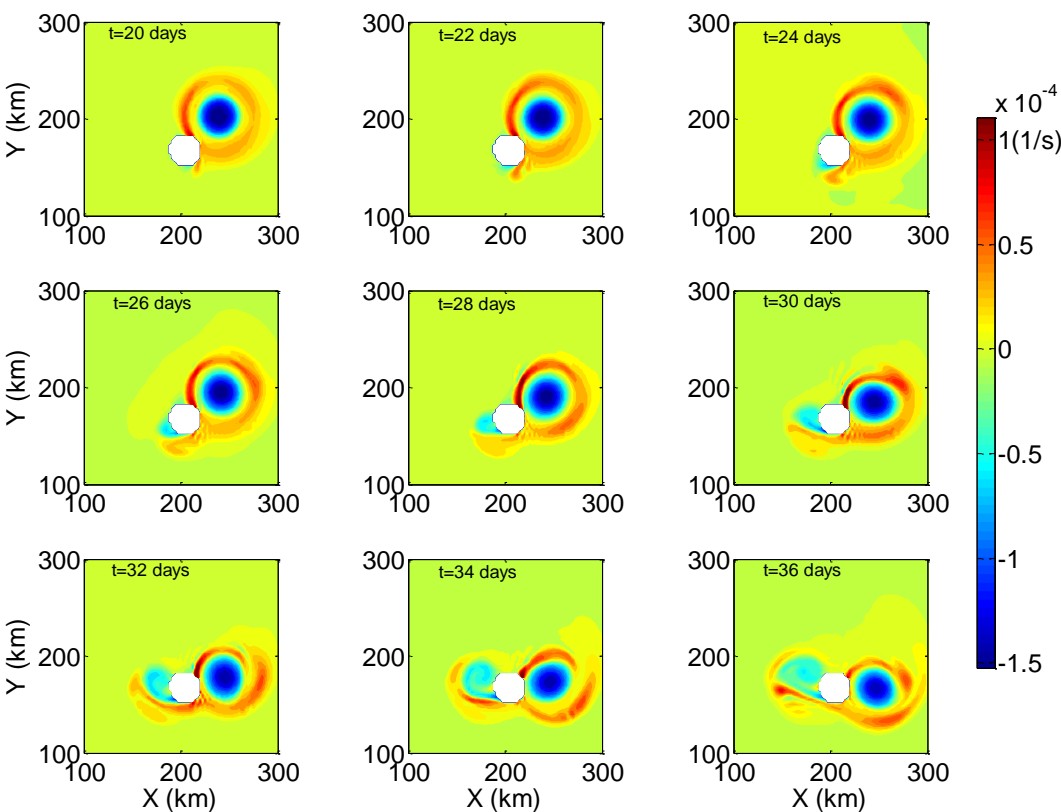

Figure 8. The temporal evolution of the eddy in the interaction with an island of 20 km in diameter. The colours represent the PVA at 100 m depth.

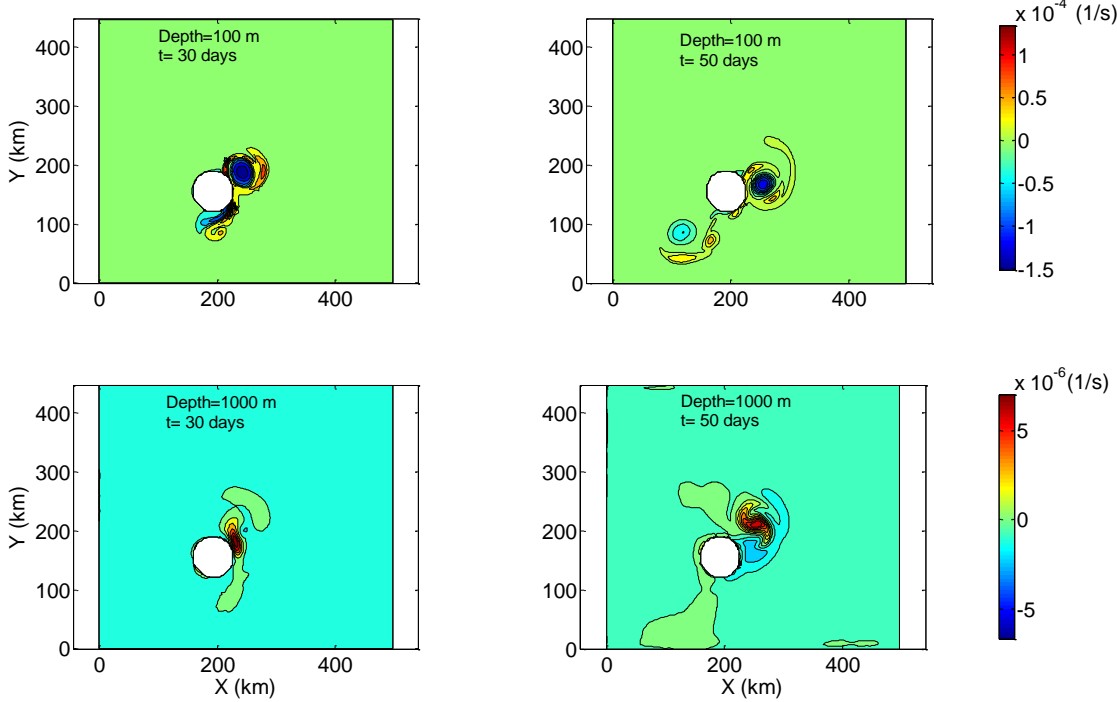

Figure 9. The potential vorticity anomaly (PVA) distributions of the interaction between the warm eddy (anticyclonic eddy) and the island of 60 km in diameter at 100 m (upper) and 1000 m (lower) at day 30 and 50. The colours representing the PVA and the solid lines being the PVA contours.

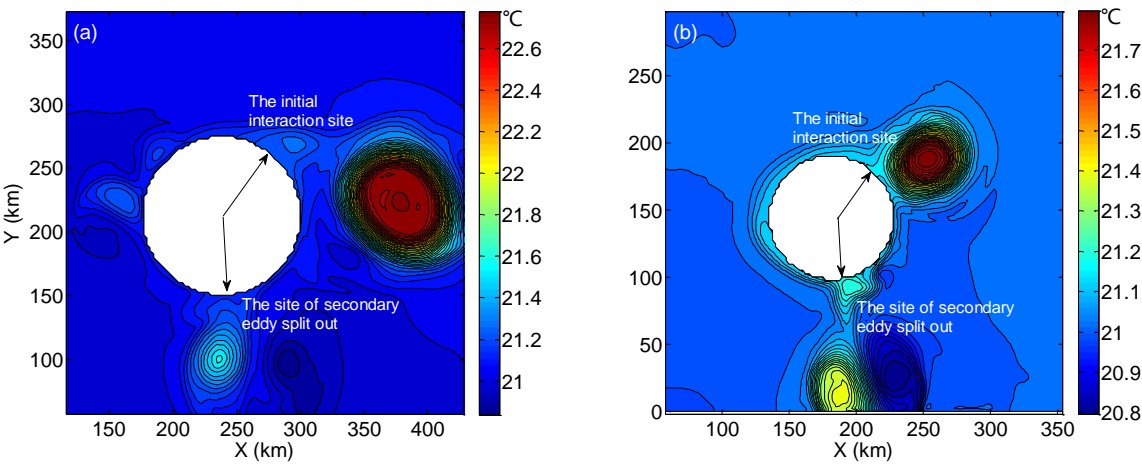

Figure 10. Comparison of the interactions between different sized islands and eddies (a) the island diameter is 120 km

and the eddy diameter is 90 km; (b) the island diameter is 90 km and the eddy diameter is 60 km. The colours represent the temperature at 100 m depth and the black arrows indicate the initial eddy-island interaction site and the site of secondary eddy split out.

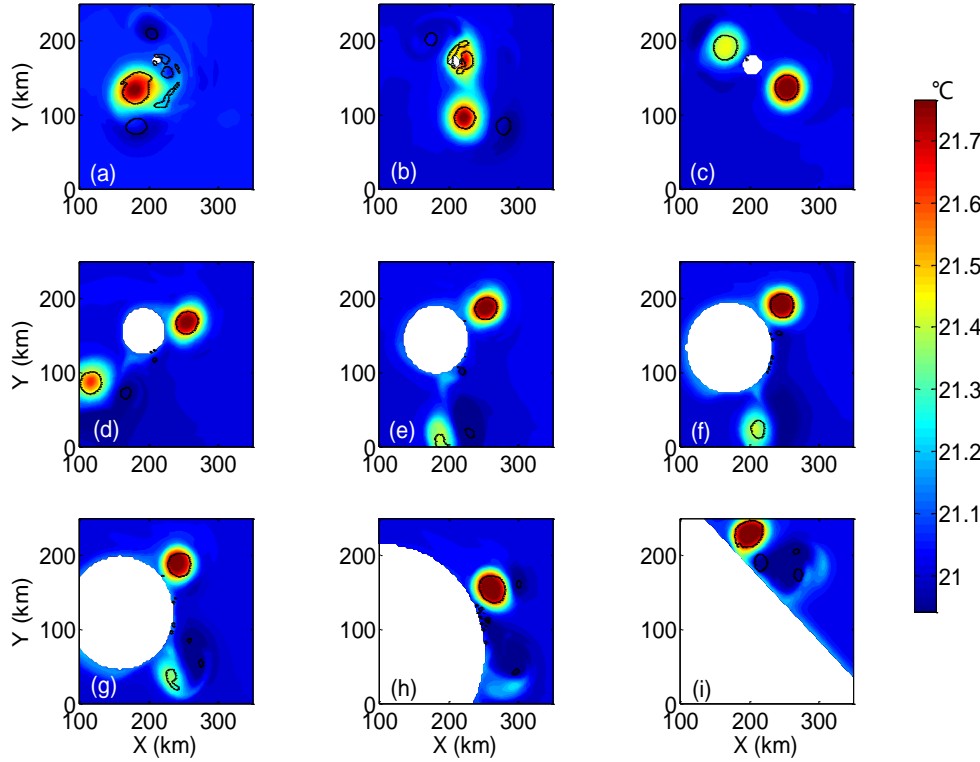

Figure 11. The results of the eddy-island interaction after 50 days for islands with different diameter (a): 10 km; (b): 15 km; (c): 25 km; (d): 60 km; (e): 90 km; (f): 120 km; (g): 150 km; (h): 300 km; (a): infinite. The colours represent temperature at 100 m depth and the black solid lines are O-W parameter with value of $-0.2\sigma_w$.

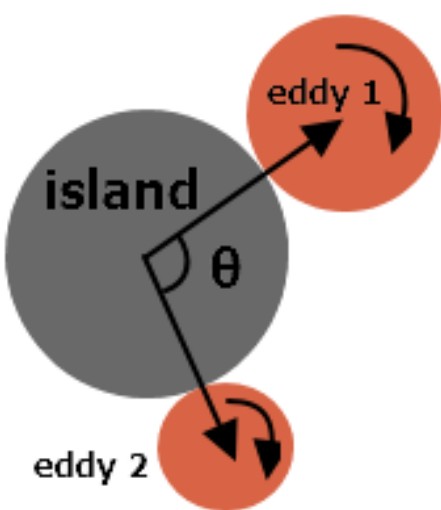

Figure 12. Sketch illustrating the position relationship of the two split eddies. When the eddy (eddy 1) encounters the island, the secondary eddy (eddy 2) splits out at angle θ during the splitting.

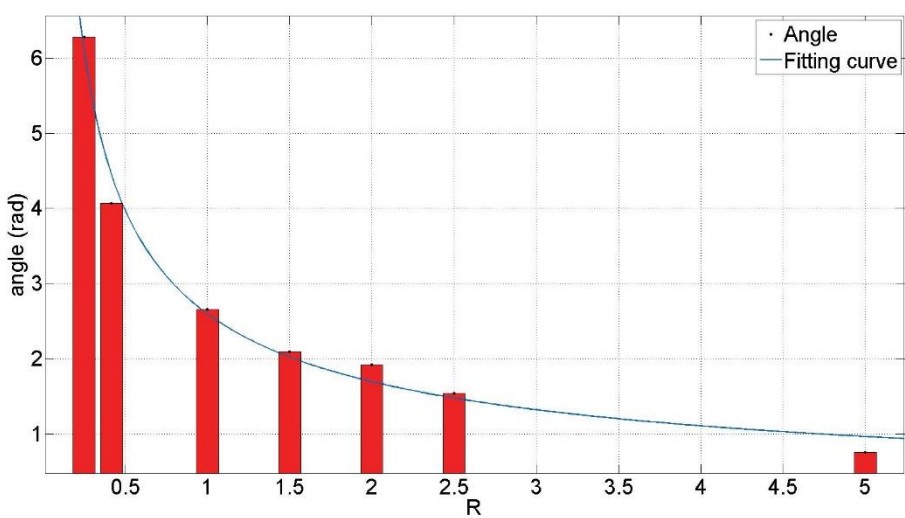

Figure 13. Distribution of relative angle (rad) with island size (R). The blue line is the fitting curve.

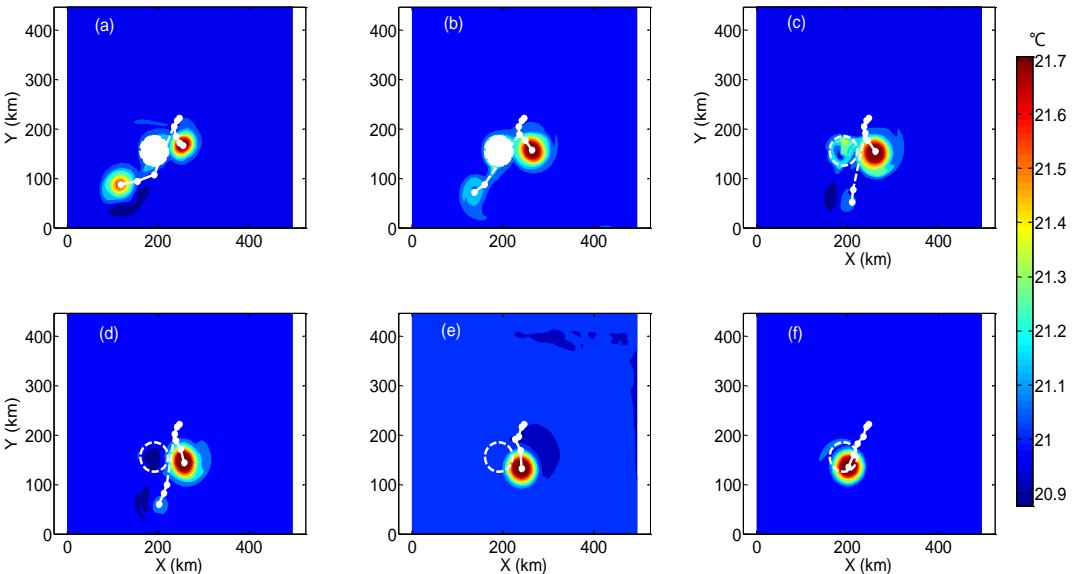

Figure 14. Eddy evolution in the case of the interaction with a seamount of 60 km in diameter at 100 m depth for different submergence depths (a): 10 m; (b): 50 m; (c): 100 m; (d): 200 m; (e): 500 m; (f):1000 m. The temperature is shown in colours, and the trajectory of the eddy center is shown in white dotted lines.

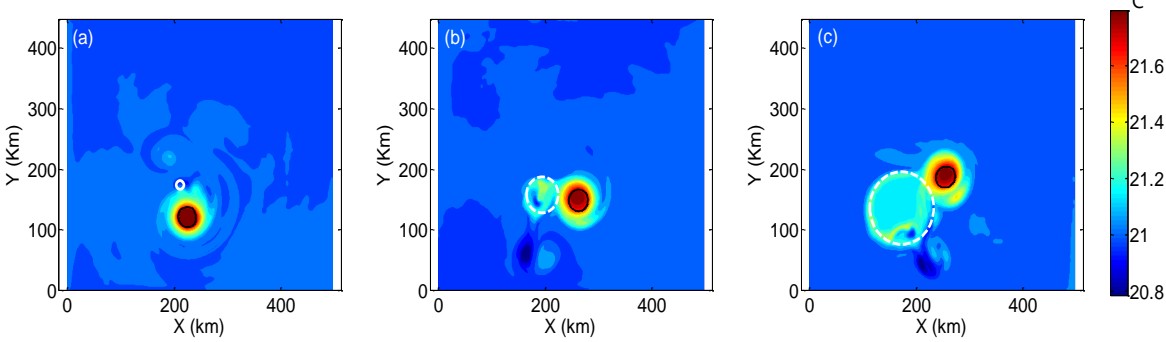

Figure 15. Eddy evolution at 100 m depth during the interaction with different size seamounts with submergence depth 100 m. (a): 15 km; (b): 60 km; (c): 120 km. The temperature is shown in colours, and the white dashed lines are the position of seamount; the black solid lines are O-W parameter with value of $-0.2\sigma_w$.

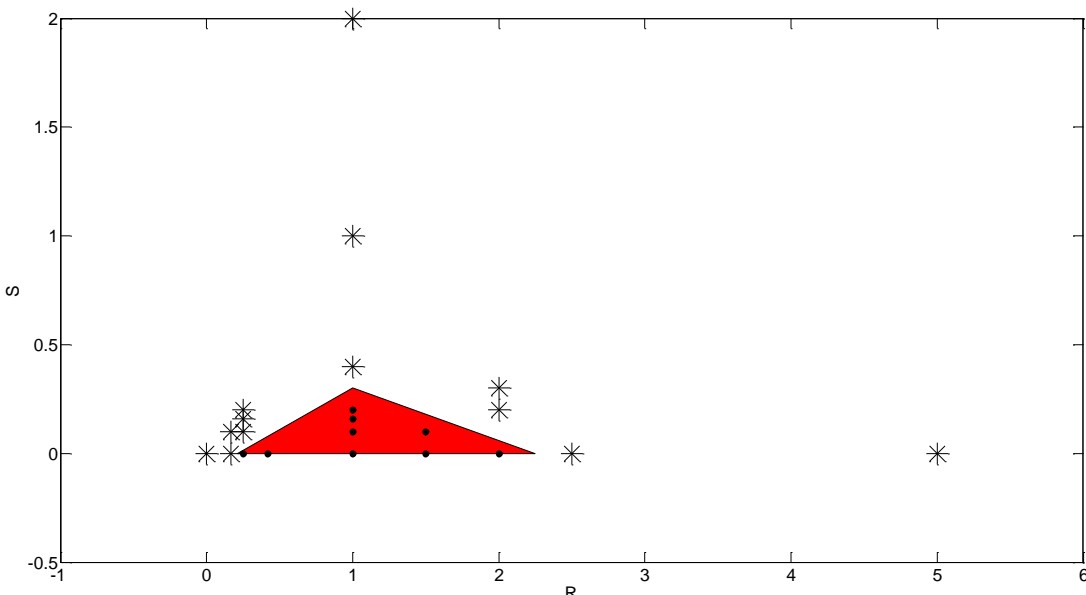

Figure 16. The range of parameters studied and dependence of qualitative features of the collision between eddies and obstacles for the range of R and S. The star marks represent no eddy splitting in the collision, and the solid dots represent eddy splitting. The red area is the range of eddy splitting.

