# Peer review of "A modelling study of eddy-splitting by an Island/Seamount"

_Ocean Science, 2016_

## Short Comment (SC1) · 12 Dec 2016

General comments

The manuscript is very interesting, and adds to the literature on interactions between eddies and topography. By introducing two non-dimensional parameters for island size and submergence depth, they identify no splitting, weak splitting, and splitting regimes. To my knowledge, this is new. Therefore, the paper should be published.

However, when reading this some questions/issues came up and the manuscript will benefit from a revision before being published, taking the comments, small and more major, into account.

Firstly, they use the MITgcm with DX = 2.5 km and DX = 50 m near the surface in their

study. They state correctly that MITgcm is non-hydrostatic. Have they used it in non-hydrostatic mode? They do not have the resolution to capture non-hydrostatic effects, and these effects may be important in the boundary layer towards the island/seamount as the eddy start to interact. There will be an upstream blocking effect and transfer of energy to smaller scales in this area. I noticed fairly large values of heat diffusion co-efficients and vertical eddy viscosity. Are these necessary to avoid instabilities around the island. It would be interesting to see figures of the temperature, vorticity, and/or flow field in the near field around the island. Are the overall rules of thumb given robust to model choices, grid size, and parameterizations?

Secondly, for uniform flow against an obstruction there will be eddies generated and oscillations. Non-hyd. pressure in the boundary layer plays a crucial role. With a sequence of eddies hitting an island, will the next eddy be shed in the same way as the first? I would think that the first eddy leaves a vorticity field behind around the island, and this will affect the next?

Thirdly, in the experiments, the island/seamount is placed in the middle of the trajectory of the eddy. Do they always hit spot on? Can it be splitting if they hit more to one side of the island?

It will be too much to address these issues in the present paper, but they may include some discussions of this.

Specific Comments

Page 7: We believe that the secondary eddy comes from eddy-splitting rather than being formed independently. Remark: This can be checked by using a tracer, and will give added value, see second remark above.

Page 8 line 10: which positive vorticity has increases Remark: Fig 6 is a snapshot after 50 days. How can we see that PV has increased?

In the interaction with seamount studies only warm eddies are used. Are the warm and

cold eddy cases symmetric?

Technical Corrections

Page 1 line 15: And the scale ... > The scale ...

Page 2 line 22: Chang et al. (Chang et al.,2012) > Chang et al.(2012) Similar remark on Kersale et al. below

Page 4 around Eq. 1: Remove . before the equation and change Where to where.

Page 4 around Eq. 2 and 3: Remove : and Where to where

Page 4 line 19: compoment > component

Page 6 line 7: warm eddy and cold eddy > the warm eddy and the cold eddy

Page 6 lines 20 and 21: But the cold eddy > However, the cold eddy

Page 7 line 7: mirror symmetry > mirror symmetric

Page 7 line 11: Diameter > The diameter

Page 8 line 3: generation > the generation

Page 8 line 4: to eddy generation > to the eddy generation

Page 8 line 4: shows the cyclone > shows that the cyclone

Page 9 line 1: Different > different

Page 9 line 3: "has the similar behavior" : Please, reformulate.

Page 9 line 6: Start the sentence: In the study, we define two ..

Page 9 line 12: shows when > shows that when

Page 11 line 10: eddy-splitting not > eddy-splitting is not

Page 11 line 18: This is reason > This is the reason

Page 11 line 21: main settings of experiments > the main settings of the experiments

---

## Author Comment (AC1) · 14 Dec 2016

We are very grateful to your time and efforts. Your insightful comments will greatly help us to improve our present manuscript and our further work. In the revised manuscript we will follow your (and others') comments to address the issues raised. Here are our replies to your comments.

General comments

(1)Have they used it in non-hydrostatic mode. . .Are the overall rules of thumb. . .

Reply: We have not used the non-hydrostatic option although the model is non-hydrostatic. As you say, the model resolution cannot capture the non-hydrostatic effect. So we turn it off to reduce computational cost. We agree with you that the nonhydrostatic effect is important in the interaction and we will try to improve our resolution of the model to study the energy transfer in our next work. As to the values of heat diffusion and vertical eddy viscosity used in model, main purpose is to avoid instabilities around island. Some limited model experiments show that the results of the eddy movement and eddy splitting are robust.

(2)With a sequence of eddies hitting an island. . .

Reply: What you have mentioned is very interesting, and we guess that a sequence of eddies hitting an island will be much different from an isolated eddy hitting an island. This will be an aspect we should explore in our next phase of the work.

(3)In the experiments, the island/seamount is placed in. . .

Reply: A related matter was mentioned in the work of Simmons and Nof (2000) who idealized the seamount as a wall (see references in the manuscript). However, large differences have been found when an eddy colliding with an island compared with a wall. So more work is needed to test the influence of hitting point in the eddy-island interaction.

Specific comments

(1)Page 7: We believe that the secondary eddy comes from eddy-splitting rather than being formed independently. Remark: This can be checked by using a tracer, and will give added value, see second remark above.

Reply: Yes, you are right. Actually the temperature used in the model can be seen as a kind of tracer. We will reformulate this sentence in the revision.

(2)Page 8 line 10: which positive vorticity has increases Remark: Fig 6 is a snapshot after 50 days. How can we see that PV has increased?

Reply: Yes, in the revision we will substitute it with a time series of snapshots of the eddy evolution to see the change of PV.

(3)In the interaction with seamount studies only warm eddies are used. Are the warm and cold eddy cases symmetric?

Reply: We will add more similar experiments using cold eddies to check whether it will be symmetric with warm eddy cases.

Technical corrections

We will correct these in the revision, many thanks.

---

## Short Comment (SC2) · 15 Dec 2016

In general, the topic is interesting and the study is relevant for Ocean Science. There will be numbers of extension works following this study, given that it clearly figures out the frame of such works. However the broad but imprecise introduction, the deficient design of study cases, and the ambiguous choice of parameters may reduce its potential value. So I suggest the authors should better clarify their frame work of studies in generally, and focus on their problem more precisely.

Major comments

The title is vague. Some critical information may be appended to the title, such as "with f-plane" or "$\beta$-plane", "Island size", "seamount submergence depth", etc.

As an ideal simulation, the authors should not be constrained by the background or previously study on the South China Sea (e.g., Wang et al., 2003, 2005). Otherwise, the authors can directly simulate the observed eddy-island interaction events near Dongsha Island.

The study is in f-plane or $\beta$-plane? Since the present study zone is relatively small, it is better to use f-plane rather than $\beta$-plane approximation for simplification. For example, supposing 100 km with $\beta=2\times10^{-11}m^{-1}s^{-1}$, the Coriolis parameter only varies about $2 \times 10^{-6}s^{-1}$, which is only 2% of the local Coriolis parameter f=$9\times10^{-5}s^{-1}$. And the eddy motion in this study has notable different from that in $\beta$-plane (e.g. Early et al., 2011). The study on eddy splitting in f-plane itself is a valuable work. This can also be a baseline for further study with $\beta$-plane approximation.

The eddy is mesoscale or sub-mesoscale? As mentioned in first sentence, the mesoscale eddies (scale of 100 km) and sub-mesoscale eddies (scale of 10 km) have different scales. Although the choice of ideal Gaussian-type profile (e.g., Zhang et al. 2013, Wang et al., 2015) is valid for this study, the choice of L=15 km is subtle. It might be better to choose a typical value either for mesoscale eddies (e.g., L=50 km) or for sub-mesoscale eddies (e.g., L=5 km) in this study.

There are lots of physical effects and control parameters in eddy-splitting due to eddy-island interaction. The authors introduce two dimensionless parameters R and S. They are important, but they may not enough. According to our previous study (unpublished work), the eddy amplitude/strength, the speed of eddy motion, and the distance between eddy and island will play the comparable role as R and S. In present study, the authors may want to give a comprehensive review of these effects, then focus their study only on one or two of them by fixed other parameters.

Minor comments

Page 2, line 1 Guihua et al. 2005 ->Wang et al., 2005. And the reference is corrected as below. And another pioneer paper (Wang et al 2003) should be better cited here.

[Figure]

Page 2, line 3, The paper in Science (Zhang et al., 2014) should be better cited here.

Page 3, line 10 "there are few cases of eddy-splitting found by satellite images so far." our recent published paper in this journal (GEM: a dynamic tracking model for mesoscale eddies in the ocean) just illustrates such kind of case.

Page 4, line 10, "Fig.1 shows the temperature and azimuthal velocity distribution on the cross section through the eddy center." Page 6, line 10 At the beginning of the model integration, the eddy will adjust itself to a dynamic balance. It is better to show the balanced eddy structure rather than the initial eddy structure in Fig.1.

Page 5, line 9, "parameter $\beta$ = 2 $\times 10^{-11} m^{-1} s^{-1}$ and the Coriolis parameter f = $9 \times 10^{-5} s^{-1}$, which are the typical values in the SCS, are used in model." This is not correct. Such values suit for $38^o N$ are taken from Wei and Wang (2009). While at Dongsha Island ($20^o N$) in SCS, the parameter $\beta$ = 2.15 $\times 10^{-11} m^{-1} s^{-1}$ and the Coriolis parameter f =$5 \times 10^{-5} s^{-1}$.

References:

Early J J, Samelson R M, Chelton D B (2011). The evolution and propagation of quasi-geostrophic ocean eddies. J Phys Oceanogr, 41 (8): 1535–1555.

Wang Z, Li Q, Sun, L, Li S., Yang Y.-J., and Liu S.S. (2015). The most typical shape of oceanic mesoscale eddies from global satellite sea level observations, Front. Earth Sci. 2015, 9 (2): 202-208. DOI 10.1007/s11707-014-0478-z.

WANG, G.H., CHEN, D. and CHU, P., 2003, Mesoscale eddies in the South China Sea observed with altimeter data. Geophysical Research Letters, 30, 2121, doi:10.1029/2003GL018532.

Wang, G.H., Su. J.L., and Li, R.F.: Mesoscale eddies in the South China Sea and their impact on temperature profiles, Acta oceanologica sinica, 24, 9-45, 2005.

Zhang Z G, Zhang Y, Wang W, Huang R X (2013). Universal structure of mesoscale

eddies in the ocean. Geophys Res Lett, 40(14): 3677–3681.

Zhang Z, Wang W, Qiu B. Oceanic mass transport by mesoscale eddies. Science, 2014, 345(6194): 322-324.

---

## Author Comment (AC2) · 19 Dec 2016

We are very grateful to your time and efforts. Your comments will greatly help us to improve our present manuscript and further work. We will consider your comments carefully and improve our manuscript following your (and others') comments.

General comments:

(1) The title is vague. . .As an ideal simulation. . .

Reply: Our study is motivated by eddy-splitting in the South China Sea. So we emphasize the South China Sea in our paper. Then we expand our experiments to different size islands and seamounts with different submergence depth. As a result, we decide our title including main components without many details. However, we will reconsider

this carefully in our revised manuscript.

(2) The study is in f-plane or $\beta$-plane...

Reply: The study is on $\beta$-plane. The $\beta$ effect is primary force in the movement of the eddy, even though the study area is relatively small. On the f-plane, without any external forcing, the eddy will not move. This is why we choose $\beta$-plane.

(3) The eddy is mesoscale or sub-mesoscale...

Reply: The eddy in the study can be regarded as mesoscale. The scale of the eddy used in the model is based on statistics properties of ACEs in the northern SCS (e.g., Nan et al., 2011; Zhang et al., 2013; Chen et al., 2010, see references in the manuscript). Though the study is an ideal simulation, we prefer a more realistic eddy (here reflected in the eddy scale) for better applications in the future.

(4) There are lots of physical effects and control parameters...

Reply: There are indeed many factors influencing eddy splitting during the eddy-island interaction. The parameters mentioned in your comments are all important. Here we only investigate fully developed eddies; therefore the topography (dimensionless parameter R and S) will be the most important factor. The other parameters will be explored in the future work.

Special comments

(1) Page 2, line 1... line3...

Reply: Thank you for your suggestion, we will add their references in the revision.

(2) Page 3, line 10...

Reply: Reference of GEM: a dynamic tracking model for mesoscale eddies in the ocean will be added.

(3) Page 4, line 10...

Reply: Changes of the structure of the eddy are insignificant before the eddy-island interaction, we, therefore, don't give the details in the manuscript.

(4) Page 5, line 9...

Reply: Thank you for your correction. The results are not sensitive to these two different values of f or $\beta$.
* * *

---

## Referee Comment (RC1) · Y. Lu (Referee) · 30 Jan 2017

General comments

This manuscript presents the results of a series idealized ocean modelling simulations that illustrate the different behavior of eddy-splitting when an island/seamount is included in model's bathymetry. While the simulation results appear to be interesting, I have the following concerns with respect to the generalization and dynamic interpretation of the results:

1. The "Introduction" section provides a review of previous studies regarding eddy behavior under the influence of topography; however, the remaining questions and challenges on "eddies under the influence of island/seamount" are not explicitly explained. P3L10 states "The special processes and characteristics of splitting have not

been elucidated completely", and P3L15 says "to examine its kinematic characters and test eddy splitting process using numerical simulations". Indeed, the above statements are consistent with what being presented: the paper primarily focuses on describing kinematics of simulated eddies but offers little on understanding dynamics. One wonders whether this is sufficient for a primary publication. 2. A major conclusion of the study is the dependence of eddy behavior on two non-dimensional numbers: R the ratio of island radius to eddy radius, and S the ratio of eddy submergence depth to eddy vertical depth. The question to ask is: can eddy radius and vertical depth be all arbitrary? What role does background stratification – that defines the local Rossby radius of deformation – play in defining these length scales? I note that the background stratification is the same for all the model experiments. Can this limit the generalization of the dependence of splitting behavior on R and S? I feel that besides simply describing kinematics, providing dynamic explanation of the model results will make this study more valuable. 3. English writing needs significant improvement.

Technical corrections:

1. Model parameters: P5L13: 10^(-4) m^2/s for diffusion of heat: it is bit large for vertical but is way too small for horizontal. 2. Reference citation: a format seems to be odd, e.g., P2L23, "Chang et al. (Chang et al., 2012)", etc. 3. P12L17: reference of Sheng and Tang (2003): this study is for the Caribbean Sea but not for SCS. 4. P13L22 "Guihua, W" should be "Wang, G.", and similar for other co-authors listed.

---

## Author Comment (AC3) · 13 Feb 2017

We are very grateful to your time and efforts. Your comments will greatly help us to improve our present manuscript and further work. We will consider your comments carefully and improve our manuscript following your (and others') comments.

General comments

(1) The "Introduction" section provides. . .The paper primarily focuses on. . .

Reply: The eddy-topography interaction is affected by many factors. This study focuses on the influences of island/seamount on eddy splitting. Your comments benefit our improvement and we will correct the deficiencies in the introduction in the revised version.

(2) A major conclusion of the study...The question to ask is...

Reply: In the section 4.3 of the paper, we compared interactions between different sized eddies and islands. When they have a similar scale ratio (R and S), they have similar eddy splitting pattern. In this way, the eddy radius and vertical depth can be arbitrary. As to background stratification, we did not discuss how it effects the behavior of eddy splitting. We think that ocean stratification is important like the other factors mentioned in the short comments. We will expand our study to explore those aspects in our next phase of the work.

(3) English writing needs significant improvement.

Reply: Thank you for your suggestions, we will do our best in the revised version.

Technical corrections

We will correct these in the revised version, many thanks.

---

## Referee Comment (RC2) · Anonymous Referee #2 · 27 Mar 2017

I have seen the responses, and they are on the right track. I would like to see that they follow up on all remarks, into a revised paper before going for accept.

---

## Author Response (AR1)

**A modelling study of eddy-splitting by an Island/Seamount**

**By Yang et al.**

**The response to the Referees' comments and other short comments**

5 We are very grateful to the time and efforts of the editor and referees. Your comments greatly help us to improve our present manuscript and further work. We have considered your comments carefully and improved our manuscript following the comments. Please note our one by one responses to your comments in blue and also revised manuscript with track changes in red attached at the end.

10 **Referee #1:**

**General comments**

This manuscript presents the results of a series idealized ocean modelling simulations that illustrate the different behavior of eddy-splitting when an island/seamount is included in model's bathymetry. While
15 the simulation results appear to be interesting, I have the following concerns with respect to the generalization and dynamic interpretation of the results:

1. The "Introduction" section provides a review of previous studies regarding eddy behavior under the influence of topography; however, the remaining questions and challenges on "eddies under the influence of island/seamount" are not explicitly explained. P3L10 states "The special processes and characteristics
20 of splitting have not been elucidated completely", and P3L15 says "to examine its kinematic characters and test eddy splitting process using numerical simulations". Indeed, the above statements are consistent with what being presented: the paper primarily focuses on describing kinematics of simulated eddies but offers little on understanding dynamics. One wonders whether this is sufficient for a primary publication.

Reply: The eddy-topography interaction is affected by many factors. This study focuses on the influences

of island/seamount on eddy splitting. Your comments benefit our improvement and we have corrected the deficiencies in the introduction, and added the comments on mechanics of eddy-splitting in the revised version which is attached at the end in section 4.2. Our quantitative results, namely, the relationship between eddy-splitting and two dimensionless parameters, have wide implications and have not been reported in the literature previously.

2. A major conclusion of the study is the dependence of eddy behavior on two non-dimensional numbers: R the ratio of island radius to eddy radius, and S the ratio of eddy submergence depth to eddy vertical depth. The question to ask is: can eddy radius and vertical depth be all arbitrary? What role does background stratification – that defines the local Rossby radius of deformation – play in defining these length scales? I note that the background stratification is the same for all the model experiments. Can this limit the generalization of the dependence of splitting behavior on R and S? I feel that besides simply describing kinematics, providing dynamic explanation of the model results will make this study more valuable.

Reply: In section 4.3 of the paper, we compared interactions between different sized eddies and islands. When they have a similar scale ratio (R and S), they have similar eddy splitting patterns. Therefore we can say that the eddy radius and vertical depth can be arbitrary.

As to background stratification, we did not discuss how it affects the behavior of eddy splitting. We think that ocean stratification is important like the other factors mentioned in the short comments. We will expand our study to explore those aspects in our next phase of the work. In addition, we add the explanation of eddy-splitting in section 4.2.

3. English writing needs significant improvement.

Reply: Thank you for your suggestions, we have done our best in the revised version following your and other referee's suggestions.

**Technical corrections:**

1. Model parameters: P5L13: 10ˆ(-4) mˆ2/s for diffusion of heat: it is bit large for vertical but is way too small for horizontal.

Reply: There was an error in the original manuscript. We used vertical eddy diffusivity of 10^(-4) m^2/s and zero horizontal diffusivity, which is now stated in the revised manuscript. The vertical diffusivity value of 10^(-4) m^2/s may be a bit large, but it compensates the zero horizontal diffusivity. The main purpose for the use of the diffusion terms is to avoid instabilities around islands without too much soomthing. We have done some tests to see the sensitivity of model results to diffusivity/viscosity parameters and found that the results are robust.

2. Reference citation: a format seems to be odd, e.g., P2L23, "Chang et al. (Chang et al., 2012)", etc.

3. P12L17: reference of Sheng and Tang (2003): this study is for the Caribbean Sea but not for SCS. 4. P13L22 "Guihua, W" should be "Wang, G.", and similar for other co-authors listed.

Reply: All technical corrections were made in the text, thanks for the detailed suggestions.

**Referee #2:**

**General comments**

The manuscript is very interesting, and adds to the literature on interactions between eddies and topography. By introducing two non-dimensional parameters for island size and submergence depth, they identify no splitting, weak splitting, and splitting regimes. To my knowledge, this is new. Therefore, the paper should be published.

However, when reading this some questions/issues came up and the manuscript will benefit from a revision before being published, taking the comments, small and more major, into account.

Firstly, they use the MITgcm with DX = 2.5 km and DX = 50 m near the surface in their study. They state correctly that MITgcm is non-hydrostatic. Have they used it in non-hydrostatic mode? They do not have

the resolution to capture non-hydrostatic effects, and these effects may be important in the boundary layer towards the island/seamount as the eddy start to interact. There will be an upstream blocking effect and transfer of energy to smaller scales in this area. I noticed fairly large values of heat diffusion coefficients and vertical eddy viscosity. Are these necessary to avoid instabilities around the island. It would be interesting to see figures of the temperature, vorticity, and/or flow field in the near field around the island. Are the overall rules of thumb given robust to model choices, grid size, and parameterizations?

Reply: We are pleased to read that 'The manuscript is very interesting, and adds to the literature on interactions between eddies and topography', and 'it is new'.

We have not used the non-hydrostatic option although the model have the capability of non-hydrostatic dynamics. As you say, the model resolution cannot capture the non-hydrostatic effect. We agree with you that the non-hydrostatic effect is important in the interaction near the boundary and energy transfer to smaller scales, which are the future topics of research. However in this paper the main focus is on the eddy evolution and eddy-splitting which we think the non-hydrostatic dynamics plays a minor role.

As to the values of the heat diffusion and vertical eddy viscosity used in the model, the main purpose is to avoid instabilities around the island. Some limited model experiments show that the results of the eddy movement and eddy splitting are robust (also see our response to referee #1).

Secondly, for uniform flow against an obstruction, there will be eddies generated and oscillations. Non-hyd. pressure in the boundary layer plays a crucial role. With a sequence of eddies hitting an island, will the next eddy be shed in the same way as the first? I would think that the first eddy leaves a vorticity field behind around the island, and this will affect the next?

Reply: What you have mentioned is very interesting, and we guess that a sequence of eddies hitting an island will be different from an isolated eddy hitting an island. This is an aspect we should explore in our next phase of the work.

Thirdly, in the experiments, the island/seamount is placed in the middle of the trajectory of the eddy. Do they always hit spot on? Can it be splitting if they hit more to one side of the island?

Reply: A related matter was mentioned in the work of Simmons and Nof (2000) who idealized the seamount as a wall (see references in the manuscript). However, large differences have been found when an eddy colliding with an island compared with a wall. So more work is needed to test the influence of the hitting point in the eddy-island interaction.

5    It will be too much to address these issues in the present paper, but they may include some discussions of this.

Reply: Thanks for your suggestions. We have added the relevant discussion on these issues in the last section of the paper.

10    **Specific Comments**

Page 7: We believe that the secondary eddy comes from eddy-splitting rather than being formed independently. Remark: This can be checked by using a tracer, and will give added value, see second remark above.

Reply: Yes, you are right. Actually, the temperature used in the model can be seen as a kind of tracer. We
15    have reformulated this sentence in the revision.

Page 8 line 10: which positive vorticity has increased Remark: Fig 6 is a snapshot after 50 days. How can we see that PV has increased?

Reply: Yes, in the revision we substitute it with a time series of snapshots of the eddy evolution to see the change of PV.

20    In the interaction with seamount studies, only warm eddies are used. Are the warm and cold eddy cases symmetric?

Reply: We have added one model experiment using a cold eddy to investigate this and found the results are similar.

25    **Technical Corrections**

Page 1 line 15: And the scale ... > The scale ...

Page 2 line 22: Chang et al. (Chang et al.,2012) > Chang et al.(2012) Similar remark on Kersale et al. below

Page 4 around Eq. 1: Remove . before the equation and change Where to where.

Page 4 around Eq. 2 and 3: Remove : and Where to where

Page 4 line 19: compoment > component

Page 6 line 7: warm eddy and cold eddy > the warm eddy and the cold eddy

Page 6 lines 20 and 21: But the cold eddy > However, the cold eddy

Page 7 line 7: mirror symmetry > mirror symmetric

Page 7 line 11: Diameter > The diameter

Page 8 line 3: generation > the generation

Page 8 line 4: to eddy generation > to the eddy generation

Page 8 line 4: shows the cyclone > shows that the cyclone

Page 9 line 1: Different > different

Page 9 line 3: "has the similar behavior" : Please, reformulate.

Page 9 line 6: Start the sentence: In the study, we define two ..

Page 9 line 12: shows when > shows that when

Page 11 line 10: eddy-splitting not > eddy-splitting is not

Page 11 line 18: This is reason > This is the reason

Page 11 line 21: main settings of experiments > the main settings of the experiments

Reply:All technical corrections were made in the text, many thanks.

**Short comments by L. Sun:**

In general, the topic is interesting and the study is relevant for Ocean Science. There will be numbers of

extension works following this study, given that it clearly figures out the frame of such works. However, the broad but imprecise introduction, the deficient design of study cases, and the ambiguous choice of parameters may reduce its potential value. So I suggest the authors should better clarify their framework of studies in generally, and focus on their problem more precisely.

5 **Major comments**

The title is vague. Some critical information may be appended to the title, such as "with f-plane" or "β-plane", "Island size", "seamount submergence depth", etc.

As an ideal simulation, the authors should not be constrained by the background or previously study on the South China Sea (e.g., Wang et al., 2003, 2005). Otherwise, the authors can directly simulate the
10 observed eddy-island interaction events near Dongsha Island.

Reply: Our study is motivated by eddy-splitting in the South China Sea. We, therefore, emphasize the South China Sea in our paper. Then we expand our experiments to different size islands and seamounts with different submergence depths. As a result, we decide this title including main components without many details.

15 The study is in f-plane or β-plane? Since the present study zone is relatively small, it is better to use f-plane rather than β-plane approximation for simplification. For example, supposing 100 km with $\beta = 2 \times 10^{-11} m^{-1} s^{-1}$, the Coriolis parameter only varies about $2 \times 10^{-6} s^{-1}$, which is only 2% of the local Coriolis parameter $f = 2 \times 10^{-5} s^{-1}$. And the eddy motion in this study has notable different from that in β-plane (e.g. Early et al., 2011). The study on eddy splitting in f-plane itself is a valuable work.
20 This can also be a baseline for further study with β-plane approximation.

Reply: The study is on β-plane. The β effect is the primary force in the movement of the eddy, even though the study area is relatively small. On the f-plane, without any external forcing, the eddy will not move. This is why we used β-plane.

The eddy is mesoscale or sub-mesoscale? As mentioned in first sentence, the mesoscale eddies (scale of
25 100 km) and sub-mesoscale eddies (scale of 10 km) have different scales. Although the choice of ideal

Gaussian-type profile (e.g., Zhang et al. 2013, Wang et al., 2015) is valid for this study, the choice of L=15 km is subtle. It might be better to choose a typical value either for mesoscale eddies (e.g., L=50 km) or for sub-mesoscale eddies (e.g., L=5 km) in this study.

Reply: The eddy in the study can be regarded as mesoscale. The scale of the eddy used in the model is based on statistics properties of ACEs in the northern SCS (e.g., Nan et al., 2011; Zhang et al., 2013; Chen et al., 2010, see references in the manuscript). Although the study is an idealized simulation, we prefer a more realistic size of the eddy (here reflected in the eddy scale) for better applications in the future.

There are lots of physical effects and control parameters in eddy-splitting due to eddy-island interaction. The authors introduce two dimensionless parameters R and S. They are important, but they may not enough. According to our previous study (unpublished work), the eddy amplitude/strength, the speed of eddy motion, and the distance between eddy and island will play the comparable role as R and S. In present study, the authors may want to give a comprehensive review of these effects, then focus their study only on one or two of them by fixed other parameters.

Reply: There are indeed many factors influencing eddy splitting during the eddy-island interaction. The parameters mentioned in your comments are all important and we have added the relevant discussion on these issues in the last section of the paper. Here we only investigate fully developed eddies; therefore the topography (dimensionless parameter R and S) will be the most important factor. The other parameters will be explored in the future work.

**Minor comments**

Page 2, line 1 Guihua et al. 2005 ->Wang et al., 2005. And the reference is corrected as below. And another pioneer paper (Wang et al 2003) should be better cited here.

Page 2, line 3, The paper in Science (Zhang et al., 2014) should be better cited here.

Page 3, line 10 "there are few cases of eddy-splitting found by satellite images so far." our recent published paper in this journal (GEM: a dynamic tracking model for mesoscale eddies in the ocean) just illustrates such kind of case.

Reply: Thank you for your suggestions, we have studied these papers carefully and added their references in the revision.

Page 4, line 10, "Fig.1 shows the temperature and azimuthal velocity distribution on the cross section through the eddy center." Page 6, line 10 At the beginning of the model integration, the eddy will adjust itself to a dynamic balance. It is better to show the balanced eddy structure rather than the initial eddy structure in Fig.1.

Reply: Changes of the structure of the eddy are insignificant before the eddy-island interaction, we, therefore, omitted the details in the manuscript.

Page 5, line 9, "parameter$\beta = 2 \times 10^{-11} m^{-1} s^{-1}$ and the Coriolis parameter $f = 9 \times 10^{-5} s^{-1}$, which are the typical values in the SCS, are used in model." This is not correct. Such values suit for 38°N are taken from Wei and Wang (2009). While at Dongsha Island (20 N) in SCS, the parameter$\beta = 2.15 \times 10^{-11} m^{-1} s^{-1}$ and the Coriolis parameter $f = 9 \times 10^{-5} s^{-1}$.

Reply: Thank you for your correction and we removed the sentence "which are the typical values in the SCS" in the revision. The results are not sensitive to these two different values of f or β.

[revised manuscript text omitted]

---

## Referee Report (RR1)

This revised manuscript has addressed most of my comments on the originally submitted version. I am fine with the scientific content now, but still see the need of improving the English. It should not be the duty of reviewers to correct the wording, so below I will just give a few examples for the authors to consider in revising the manuscript. On the other hand, I will have no issue if the Editor of the Journal sees the English presentation to be acceptable.

1. Abstract: L 8 "in the South China Sea" should be "under the condition of South China Sea"; "MITgcm" should be spelled out; L9 "The speed of" should be "The propagation speed of". L10 "some eddies may split" should be "under certain conditions the eddy may split"; L12 "sensitivity experiments of …. indicate that" should be "sensitivity experiments with varying island or seamount geometry indicate that"; L16 "seamount depth" should be "seamount submergence depth";
2. Introduction: There seems to be a lot jumping back and forth in this section between describing general characteristics of eddies and the particular condition in SCS. Can the authors consider re-organizing the sequence of presentation, such that the general characteristics are described first, followed by describing the SCS: which aspects are consistent with the general and which aspects are special? Alternatively, you can start with the first sentence "Eddies are common…", then in the second sentence "Eddies are frequently observed in the South China Sea", followed by describing previous studies on eddies in SCS, and one the way pointing out which aspects of SCS eddies are consistent with general characteristics (citing relevant papers), and which aspects are special.
3. P4L7 "abstracted to"?
4. P5L21 "geostrophic balance" was already mentioned at P4L11. Please try to avoid repeating.
5. L7P3-4, seems repeating P6L17.
6. P11L6 "Above all"?
7. P12L12 "by defining two dimensional parameters" is in fact "… the dependence on … can be summarized using two dimensionless parameters".
8. L13P7, "Y. Liu" either deleting it or changing to "Y. Lu".
9. Fig.2 caption: insufficient information. Trajectory of 50 days but also the snapshot at day 50? Temperature at what depth? I suggest checking all the figure captions!

---

## Author Response (AR2)

**A modelling study of eddy-splitting by an Island/Seamount**

**By Yang et al.**

**The response to the editor's and Referees' comments**

5    We are very grateful to the time and efforts of the editor and referees. Your comments greatly help us to improve our present manuscript and further work. We have considered your comments carefully and improved our manuscript following the comments. Please note our one by one response to your comments in blue and also revised manuscript with track changes in red attached at the end.

10   **Topic Editor:**

,by Eric J.M. Delhez.

**Comments**

As you can read, the referees who have examined the revised version of your manuscript are positive about the changes introduced with respect to the initial submission but still raise a couple of issues.

15   First of all, two of them are of the opinion that your case studies should be analyzed more extensively. They even believe that it is difficult to discuss all the details in a single paper and that some of the issues deserve a special focus in a separate manuscript. I understand that this is perhaps not the option that you would like to take. But, if you do not consider splitting your manuscript in two parts, it would be appropriate to, at least, sketch the possible issues that deserve such an in depth treatment.

20   Reply: Thank you for the time and efforts to review the manuscript again. We consider your and the referees' suggestions and comments carefully and do the corrections. In this version, In order to better understand the mechanism of the eddy-splitting, the case of the eddy interacting with the island with 20 km diameter was studied in more detail. The temperature and PVA fields in the eddy-island interaction process were analyzed. We provide a time series of snapshots of the temperature and PVA between t=20

days and t=40 days with a smaller time interval to show more details of the eddy evolution (figures 7-8 in the manuscript below) and the related statement is presented on pages 8-9.

Second, the quality of the presentation and, in particular, of the English, should be improved.

Reply: For the deficiencies in the manuscript, we do our best to correct it in this revised version. We re-organized the introduction and add the content about the relevant results of previous research work. Technical corrections (language expressions in text and figure captions) are made in this version.

**Referee #1:**

,by Youyu Lu.

**Comments**

This revised manuscript has addressed most of my comments on the originally submitted version. I am fine with the scientific content now, but still see the need of improving the English. It should not be the duty of reviewers to correct the wording, so below I will just give a few examples for the authors to consider in revising the manuscript. On the other hand, I will have no issue if the Editor of the Journal sees the English presentation to be acceptable.

1.  Abstract: L 8 "in the South China Sea" should be "under the condition of South China Sea"; "MITgcm" should be spelled out; L9 "The speed of" should be "The propagation speed of". L10 "some eddies may split" should be "under certain conditions the eddy may split"; L12 "sensitivity experiments of …. indicate that" should be "sensitivity experiments with varying island or seamount geometry indicate that"; L16 "seamount depth" should be "seamount submergence depth";

Reply: "in the South China Sea" means the research region and we think it is better than "under the condition of South China Sea" in the expression of the sentence. Therefore, it is kept in the text. Other corrections are made in the new manuscript. Thank you.

2. Introduction: There seems to be a lot jumping back and forth in this section between describing general characteristics of eddies and the particular condition in SCS. Can the authors consider re-organizing the sequence of presentation, such that the general characteristics are described first, followed by describing the SCS: which aspects are consistent with the general and which aspects are special? Alternatively, you can start with the first sentence "Eddies are common…", then in the second sentence "Eddies are frequently observed in the South China Sea", followed by describing previous studies on eddies in SCS, and one the way pointing out which aspects of SCS eddies are consistent with general characteristics (citing relevant papers), and which aspects are special.

Reply: Your suggestions made us much more profitable in writing. We have re-organized the introduction and corrected the deficiencies in the language in the manuscript.

3. P4L7 "abstracted to"?

4. P5L21 "geostrophic balance" was already mentioned at P4L11. Please try to avoid repeating.

5. L7P3-4, seems repeating P6L17.

6. P11L6 "Above all"?

7. P12L12 "by defining two dimensional parameters" is in fact "… the dependence on … can be summarized using two dimensionless parameters".

8. L13P7, "Y. Liu" either deleting it or changing to "Y. Lu".

9. Fig.2 caption: insufficient information. Trajectory of 50 days but also the snapshot at day 50? Temperature at what depth? I suggest checking all the figure captions!

Reply: All corrections are made following your suggestions. Thank you. .

**Referee #2:**

,by Jarle Berntsen.

**General comments**

This is the second version of the manuscript that I review. In my first review, I stated that "The manuscript is very interesting, and adds to the literature on interactions between eddies and topography. By introducing two non-dimensional parameters for island size and submergence depth, they identify no splitting, weak splitting, and splitting regimes. To my knowledge, this is new. Therefore, the paper should be published."

The authors have improved the quality of the manuscript based on the inputs from the reviewers. The improvements/changes of the paper are, however, minor. There is for instance one additional Figure and the others are as before. I still feel that there are issues that they could have addressed 'deeper', and there are still language issues.

Reply: Thank you for the time and efforts to review the manuscript again. Your comments pointed out some issues and problems in the manuscript which helped us to improve the quality of our work. We do our best in the revised version following your and other referee's suggestions.

With the model system they have, they could have addressed in more detail the dynamical situation when the eddy approaches and interacts with island/seamount.

Reply: In this version, the temperature and PVA field in the eddy-island interaction process were analyzed. In order to show more details of the eddy evolution, we provided a time series of snapshots of the temperature and PVA between t=20 days and t=40 days with a smaller time interval to show more details of the eddy evolution (see pages 8-9 and figures7-8 in the manuscript below).

In the discussion on page 8, near the end of section 4.2, they refer to conservation of integrated angular momentum and show eddies after 30 and 50 days in Figure 7. The scales used are different. Are the strength of the eddies the same? Is angular momentum conserved? We see two snap-shots, but not enough to see the evolution.

Reply: we are sorry for the confusion caused by different scales used in Figure 7. In order to show the

change of integrated angular momentum before and after the eddy interacts with island, we show the PVA at surface layer (Depth=100 m) and deep layer (Depth=1000 m) at t=30 days and t=50days with the same scale. Compared with the PVA field at t=30 days, the maximum upper layer anticyclonic PVA decreases at t=50 days because of the splitting while maximum lower layer cyclonic PVA increases (we substitute figure 7 with figure.9 in the manuscript below). So the strength of the eddies changes during the interaction. For a baroclinic eddy, the integrated angular momentum can be transported vertically and laterally with the surrounding water (the vertical transport of IAM is more important for anticyclonic splitting) and conservation of integrated angular momentum is not a limitation for its splitting.

In the Figure text of Figure 9 they say that we see 'Eddy evolution'. What we see is the temperature after 50 days for island with different diameter, and not the evolution over time for the specific cases. They could at least studied in more detail one case, for instance the cases shown in Figs. 6 or 7.

Reply: There was a mistake in figure caption of Fig 9, apologize. The figure shows the results of the temperature after 50 days for islands with different diameter and it is not the eddy evolution. We checked all the figure captions in case similar mistakes.

Following your suggestion, we studied the eddy evolution process in the case of the eddy interacting with the island with 20 km diameter. In order to show the details clearly, a smaller time interval (2 days) was used (see pages 8-9 and figures 7-8 in the manuscript below).

In section 4.2 in the first paragraph they for instance say that 'the eddy looses mass' and that 'separated water is dammed at the downstream of the jet'. These statements could have been followed up with analysis of the numerical results and figures to support them.

Reply: in this version, the statements on the eddy-island interaction is improved. We studied the details in the case of the island with 20 km diameter, and the evolution of temperature and PVA fields over time are shown in figures 7-8.

**Specific Comments**

Page 3 lines 6 and 7: However large > However, large
Page 3 line 13: splitting. But > splitting, but
Page 4 line3: Much previous > Previous
On page 5 I find units given in math mode rather than in usual text mode. Check throughout.
Page 5 line 20: affect model results > affect the model results
Page 10 lines 4 and 5: No need to explain R again.
Page 10 line 7: part of topography > part of the topography
Page 11 line 7: result of numerical > results of the numerical
Page 11 line 23: settings of experiments > settings of the experiments
Page 12 line 19: and combine with > in combination with
Page 12 line 22: eddy interacts > eddy interacting
Page 12 line 24: eddy. Results > eddy. The results
Page 13 line 1 and 2: Please rewrite the first sentence starting 'In short'

Reply: All technical corrections were made in the revison, many thanks.

**Referee #3:**

,by Anonymous Referee.

**General comments**

The manuscript describes a set of numerical simulations to quantify effects of cylindrical islands/seamounts on the beta-drift of surface-intensified eddies over a flat bottom. Apparently, the most striking result is the development of secondary eddies which the authors call "eddy-splitting". The parameter range of eddy-splitting is characterized by two nondimentional parameters related to the eddy size, the obstacle size and submerged depth. This interesting study can be substantially improved if the authors attempt to clarify physical mechanisms behind the eddy-splitting in relation to previous studies.

Generally, the evolution of rotating stratified fluid near vertical boundaries can be decomposed into three major components: balanced part related to the PV redistribution, internal boundary Kelvin wave, and unbalanced gravity waves (the later is likely unimportant here, e.g., Reznik and Sutyrin, J. Fluid Mech, v.527, p.235, 2005). For better understanding of the vortex interaction with boundaries, It'd be illuminating to show baroclinic PV structure on the beta-plane before the interaction (upper anticyclonic

anomaly and lower cyclonic anomaly with maximum at some depth, like in fig. 7, cf. Herbette et al. 2005). The interaction of such dipolar (hetonic) structure with a cylindrical obstacle may result in splitting both cyclonic and anticyclonic PV anomalies and formation of self-propagation structures like illustrated in fig. 8 after the interaction. This process was described using more simple QG model without Kelvin waves (Wang and Dewar, J. Phys. Oceanogr., v.33, p.2446, 2003). Some evidence of Kelvin waves propagating clockwise around the island can be seen in fig. 4 c, d. It'd be useful to show the evolution of both upper and lower temperature fields in more details, e.g., between t = 10 and t = 40 with smaller interval to see how the Kelvin waves trap and transport water. Can the site of the secondary dipole separation from the island be related to the speed of Kelvin wave (which does not depend of the island size) ?

Reply: Thank you for the time and efforts to review the manuscript. You mentioned three major components for the evolution of rotating stratified fluid near vertical boundaries which benefit us a lot. Here following your suggestions, we use the potential vorticity anomaly (PVA) to interpret the evolution and erosion of the initial vortex. The eddy evolution process in the case of the eddy interacting with the island with 20 km diameter was explored. A time series of snapshots of the temperature and PVA field between t=20 days and t=40 days are shown in the manuscript (see pages 8-9 and figures 7-8 in the manuscript below). In order to show the details of the eddy evolution clearly, a smaller time interval (2 days) is used. We find that the shear effect of the jet which generated by eddy-island collisions and the scale of the island are two main factors counting for the formation of the new anticyclonic eddy rather than Kelvin waves. There are indeed waves generated by the eddy-island collisions, and nonlinear kelvin wave are often excited during the geostrophic adjustment, but it is not reflected obviously in the model results.

In my opinion, the physical mechanisms of the secondary eddies formation in the case of an island are complicated enough and deserve to be described in a separate manuscript, part I. Effects of additional PV anomaly on the top of submerged seamount could be described in more details in Part 2 comparing

with existing dynamical framework (Sutyrin et al., Geophys. Astrophys. Fluid Dyn, v.105, p. 478, 2011).

Reply: As you said, the interactions of eddy-island and eddy-seamount are both complicated processes. Therefore, we describe the characters of the eddy-splitting and summarize the parameter range of eddy-splitting by two non-dimensional parameters related to the eddy size, the obstacle size and submerged depth. As you suggested, we paid our attention mainly on the eddy-splitting by the island in this paper. We did not focus on the dynamic details in the eddy-seamount interaction, which is a work that deserves further study. The effects of additional PV anomaly on the top of submerged seamount will be explored in our next phase of the work.

In summary, the manuscript is not free of errors in logic; alternative explanations are not explored as appropriate; biases, limitations, and assumptions are not clearly stated; previous work and current understanding is not cited and represented correctly; information is not conveyed clearly enough to be understood by the typical reader. Therefore, it needs substantial revision to make it suitable for publishing in the OS.

Reply: For the deficiencies in the manuscript, we do our best to correct it in the revised version. We re-organized the introduction and add the content about the related research results of previous work. In the eddy-slitting part, in order to show eddy evolution clearly, the temperature and PVA are analyzed with a smaller time interval (2 days) for one eddy-island interaction case. Technical corrections (language expressions in text and figure captions) are made in this version.

Our thanks for the editor and referees again.

[revised manuscript text omitted]

---

## Author Response (AR3)

**A modelling study of eddy-splitting by an Island/Seamount**

**By Yang et al.**

**The response to the editor's and Referees' comments**

We are very grateful to the time and efforts of the editor and referees. We have considered your comments carefully and improved our manuscript following the comments and suggestions. Please note our one by one response to your comments in blue and also revised manuscript with track changes in red attached at the end.

**Topic Editor:**

,by Eric J.M. Delhez.

**Comments**

The referees who examined your manuscript recommend its publication in Ocean Science but have minor comments and suggestions that could improve the quality of your work. I invite you to implement the corresponding changes before final acceptance of your manuscript.

Reply: Thank you for the time and efforts to review the manuscript again. We have considered your and the referees' suggestions and comments carefully, and done the corrections.

20 **Referee #2:**

,by Berntsen Jarle.

**General comments**

This is the third version of the manuscript that I review. In my first review, I stated that "The manuscript is very interesting, and adds to the literature on interactions between eddies and topography. By

introducing two non-dimensional parameters for island size and submergence depth, they identify no splitting, weak splitting, and splitting regimes. To my knowledge, this is new. Therefore, the paper should be published."

In the revision process, the quality of the manuscript has improved. Most of the issues raised are adequately addressed, even if there are aspects that could have been investigated further. This is always the case. In the last revision, the addition of Figs. 7 and 8 gave further insight in the evolution of the eddy interaction with the island.

I do not think that more iterations on the manuscript will improve the quality significantly. We are close to 'accept', but a minor revision to check the manuscript carefully may be necessary.

Reply: Many thanks to review the manuscript three times. Your comments helped us to improve the quality of our work greatly.

In that process, the following remarks should be considered:

On Figs. 4 and 7: They both show the evolution of eddy-splitting induced by interaction with an island of 20 km. Fig 7 give more details from days 20 to 36. I expected the results after 20 and 30 days, given in both Figs. to be equal. They are not. The reason is probably that the initial position/starting point of the eddies in the two Figs. are different. This should be clarified if the Figs are kept as is. Why is a different initial state used for the results in Fig. 7? I may have missed something here?

Reply: The results after 20 and 30 days given in both Figs. 4 and 7 are equal. They are the results of the same one experiment and share the same initial conditions. The differences are the Fig.4 showing the whole domain (X=0~500 km, Y=0~450 km), the colours without contour lines representing the temperature, and the black solid lines being O-W parameter with value of $-0.2\sigma_w$. In Fig.7, to clear the eddy-splitting process, we focus on the region of (X=100~300 km, Y=100~300 km). Here, only temperatures are shown in the figures in colours, and black solid lines are temperature contours. We added more texts in the caption in the revised manuscript.

In Fig. 9 there are 4 plots. Only a) and b) are mentioned in the caption. They give PV anomaly at 1000 m depth. In the two lower plots 1000 m depth is indicated, but I see no anomalies in these plots. Fig. 9 with caption need to be re-done before publication.

Reply: Sorry the figure is not clear. Because the values of the PVA at 1000 m depth are relative smaller than that at 100 m depth, the anomalies at 1000 m depth are not clear under the same scale. We have re-drawn this figure and re-written the captions in the revised manuscript.

Page 2 line 21: In contrast to with a > In contrast to the case with a

Page 3 line 18: affects coastal ocean > affects the coastal ocean

Page 9 line 11: ones is inner > one is the inner

Page 9 line 11: is outer > is the outer (same also on line 12)

Page 9 line 20: Fig. 11 is referenced here, before Fig. 9 is discussed. get this

in correct order.

Page 17 line 20: Chinese signs are used: Please translate to English.

Reply: All corrections have been made in the text, many thanks.

**Referee #3:**

Report: 05 Sep 2017,by Anonymous Referee.

**General comments**

In this version, the formation of the offspring eddy is illustrated with more often output of T and PVA in Fig. 7-8, so that the physical mechanisms could be better interpreted by readers.

Reply: Thank you for the time and efforts to review the manuscript again. Your suggestions and comments have benefited us greatly.

Page 1, lines 11-12: Instead of "Eddy-splitting is related to the size of the island and the submergence depth of the seamount". it'd be more accurate to say that the focus was only on such parameters as R and S.

Page 2, lines 13-15: it'd be better to keep here only refs related to continental shelf slope and move refs related to island/seamounts below.

Page 2, line 17: it'd be useful to mention after (Nof 1988) that the generation of along-wall jets can be related to nonlinear Kelvin waves (Shi and Nof 1994, Reznik and Sutyrin 2005).

5    Reply: All corrections have been made in the text, many thanks.

The ref list has to be checked, e.g., Wang and Dewar 2003, Sutyrin et al 2011 are not in the list.

Reply: Sorry for the missing of some references in the list. We have checked the list again and done the corrections, many thanks.

[revised manuscript text omitted]